# Validation of Food Compass with a healthy diet, cardiometabolic health, and mortality among U.S. adults, 1999–2018

Meghan O'Hearn ®[1] ✉, Joshua Erndt-Marino[2], Suzannah Gerber[1], Brianna N. Lauren[1], Christina Economos[1], John B. Wong ®[3,4], Jeffrey B. Blumberg ®[1] & Dariush Mozaffarian ®[1,3]

The Food Compass is a nutrient profiling system (NPS) to characterize the healthfulness of diverse foods, beverages and meals. In a nationally representative cohort of 47,999 U.S. adults, we validated a person's individual Food Compass Score (i.FCS), ranging from 1 (least healthful) to 100 (most healthful) based on cumulative scores of items consumed, against: (a) the Healthy Eating Index (HEI) 2015; (b) clinical risk factors and health conditions; and (c) all-cause mortality. Nationally, the mean (SD) of i.FCS was 35.5 (10.9). i.FCS correlated highly with HEI-2015 (R = 0.81). After multivariable-adjustment, each one SD (10.9 point) higher i.FCS associated with more favorable BMI (−0.60 kg/m$^2$ [−0.70,−0.51]), systolic blood pressure (−0.69 mmHg [−0.91,−0.48]), diastolic blood pressure (−0.49 mmHg [−0.66,−0.32]), LDL-C (−2.01 mg/dl [−2.63,−1.40]), HDL-C (1.65 mg/d [1.44,1.85]), HbA1c (−0.02% [−0.03,−0.01]), and fasting plasma glucose (−0.44 mg/dL [−0.74,−0.15]); lower prevalence of metabolic syndrome (OR = 0.85 [0.82,0.88]), CVD (0.92 [0.88,0.96]), cancer (0.95 [0.91,0.99]), and lung disease (0.92 [0.88,0.96]); and higher prevalence of optimal cardiometabolic health (1.24 [1.16,1.32]). i.FCS also associated with lower all-cause mortality (HR = 0.93 [0.89,0.96]). Findings were similar by age, sex, race/ethnicity, education, income, and BMI. These findings support validity of Food Compass as a tool to guide public health and private sector strategies to identify and encourage healthier eating.

Uncertainty and inconsistency around selecting healthier food products contribute to consumer confusion, inconsistent and suboptimal government food policy, varying industry goals regarding reformulation, and lack of nutrition-based impact investing[1–4]. To address these challenges, nutrient profiling systems (NPS) aim to provide a scientifically grounded means for evaluating the overall healthfulness of food and beverage products. Momentum around NPS is growing globally,

including for front-of-package (FOP) and product warning labeling, school meal procurement, taxation and subsidy programs, food reformulation by industry, company rankings, and more. The European Union, Australia, New Zealand, and other governments are promoting FOP labeling requirements[5,6]; retailers are using NPS to guide consumer choices (e.g., Hannaford's *Guiding Stars*, Walmart's *Great For You*, Kroger's *Opt Up*)[7–9]; and industry is voluntarily self-

[1]Friedman School of Nutrition Science and Policy, Tufts University, Boston, MA, USA. [2]Bespoke Analytics, LLC, St. Petersburg, FL, USA. [3]Tufts Medical Center, Tufts University School of Medicine, Boston, MA, USA. [4]Tufts Medical Center, Institute for Clinical Research and Health Policy Studies, Boston, MA, USA. ✉e-mail: meghan.o_hearn@tufts.edu

labeling (i.e., Danone adopted NutriScore for FOP)[10]. In addition, organizations like Access to Nutrition Index (ATNI) are using NPS to rate and rank food manufacturers and retailers globally[11]; while investors are pledging to hold their food sector investees accountable to using an NPS to report on healthfulness of product sales and product portfolios[12]. Clearly, the availability of accurate, objective, and validated NPS is of major importance for advancing nutrition and health.

Current major NPS have important limitations, including evaluating a limited range of nutritional attributes; focusing on certain attributes (e.g., total fat) inconsistent with updated scientific evidence; omitting other attributes of importance to health (e.g., diverse food ingredients, bioactive phytochemicals, processing characteristics)[1,13,14]; and scoring products by weight, volume, or serving size, creating inconsistencies depending on water and fat content[1]. For most current NPS, these limitations also result in untenable scoring of certain major food groups, requiring differing, tailored scoring principles and algorithms for different food and beverage categories, which increases subjectivity as well as contradictions for scoring mixed dishes and meals.

The Food Compass is a novel NPS that incorporates a range of 54 potentially protective and harmful nutrients, ingredients, bioactives, additives, and processing attributes, grouped across 9 domains, and selected and weighted based on the latest evidence about their relative healthfulness[15]. The Food Compass Score (FCS), ranging from 1 (least healthful) to 100 (most healthful), enables a more consistent and universal algorithm across all food and beverage categories; permits similar scoring for mixed dishes and meals; and improves convergent and discriminatory validity of product scoring compared to other major NPS[15]. However, the validity of extending FCSs of individual food product to the daily dietary patterns of individuals, and the corresponding validity for the association between a person's FCS with a validated healthy dietary pattern and major health outcomes, have not been established. Such research is critical to substantiate the utility of Food Compass as a tool for identifying healthful foods for consumers, food businesses, investors, and policy stakeholders.

In this work, we found that a person's FCS, based on the individual food and beverages they consume, was favorably associated with three validation endpoints: (a) the Healthy Eating Index (HEI) 2015, an established measure of a healthy dietary pattern; (b) the prevalence of major clinical risk factors and health conditions; and (c) the prospective risk of all-cause mortality. We also explored how each of the 9 scoring domains of the Food Compass separately related to these outcomes.

## Results

### Participant characteristics
We evaluated a nationally representative sample of 47,999 adults aged 20–85 years enrolled in the 10 most recent cycles of NHANES, 1999–2000 through 2017–2018 (Fig. S1). The mean (SD) age was 47.2 y (17.1), 52.2% were female, and 27.8% had a college degree or more (Table 1). Mean (SD) BMI was overweight (28.8 kg/m² [6.8]); blood glucose levels, HbA1c: 5.6% (0.9); fasting plasma glucose: 105.8 mg/dL (30.8); total: HDL-C ratio, high (4.0 [1.5]), and blood pressure levels, systolic: 122.8 mmHg (18.0); diastolic: 70.9 mmHg (12.5). About 42.0% of U.S. adults had metabolic syndrome; 12.9%, diabetes; 7.7%, clinical CVD; 18.9%, lung disease; and 9.8%, cancer. Only 7.4% had optimal cardiometabolic health.

### Individual Food Compass Scores
To extend FCS for specific products to an individual's overall diet, the FCS for each item reported in a person's diet was weighted by its percent contribution to that person's total energy intake, and then summed, to calculate an individual's Food Compass Score (i.FCS, theoretical range: 1 to 100). Among U.S. adults, the mean (SD) i.FCS was 35.5 (10.9), with 5th and 95th values of 19.5 and 55.3 (Fig. 1). Nearly

all (99.5%) U.S. adults had an i.FCS below 70 (a threshold previously defined at the product level for foods to encourage); and 32.7%, an i.FCS of 30 or below (a threshold previously defined at the product level for foods to minimize).

Characteristics of U.S. adults across i.FCS quintiles are presented in Table S4. Those with higher scores tended to be older, female, and have higher education and income. In crude (unadjusted) analyses, adults with a higher i.FCS were more likely to be of Asian or other racial descent and less likely to be of Non-Hispanic Black race/ethnicity; tended to have lower total energy intake; and were more likely to be non-smokers, drink more alcohol, and be less physically active. However, after adjusting these crude associations by age and sex, the differences by i.FCS quintiles in total energy intake and physical activity were attenuated.

Individuals with higher i.FCS (≥70) consumed a greater number of and percentage total energy contribution from products with FCS ≥ 70 (median [IQR] count: 13 [8, 20]; percentage energy: 65.6% [57.9, 71.4%])) compared to products with FCS 31–69 (6 [3, 9]; 14.2% [7.3, 23.2]) or FCS ≤ 30 (2 [1, 4]; 2.3% [0.2, 6.1]) (Table S5). In comparison, individuals with lower i.FCS (≤ 30) consumed a greater number of and percentage energy contribution from products with FCS ≤ 30 (11 [7, 15]; 65.0% [56.1, 74.4]) compared to products with FCS 31–69 (8 [5, 12]; 39.1% [28.3, 48.8]) or FCS ≥ 70 (2 [1,4]; 5.9% [2.1, 10.4]). At the same time, however, for any i.FCS, there was also substantial heterogeneity in the counts and energy contribution from products of different FCS (Fig. 2), indicating that different people could arrive at similar overall i.FCS in different ways.

### Validation of Food Compass against a healthy dietary pattern
To understand how a person's i.FCS, derived from the product scores for the individual food and beverage products they consume, correlated with an externally derived healthy dietary pattern, we calculated the Healthy Eating Index (HEI) 2015 for each individual, a measure of adherence to the Dietary Guidelines for Americans that has been widely validated against health outcomes including mortality, cardiovascular and metabolic endpoints, and cancer[16–19]. The i.FCS for individuals highly correlated with HEI-2015 (R = 0.81) (Fig. 3), with strong correlations persisting across population subgroups ranging from 0.76 to 0.83 (Table S6). Individually, the 9 domains of the FCS were not as highly correlated, ranging from 0.23 for i.Specific Lipids to 0.76 for i.Nutrient Ratios (Fig. S2).

### Validation of Food Compass against clinical risk factors
To understand how a person's i.FCS was cross-sectionally associated with major clinical risk factors in NHANES (body mass index [BMI] [weight in kilograms divided by height in meters squared]; systolic and diastolic blood pressure [mm Hg]; HDL-cholesterol, LDL-cholesterol, triglycerides [mg/dL], and total cholesterol: HDL ratio; and Hba1c [%] and fasting plasma glucose [mg/dL]), we evaluated survey-weighted linear regression models for each standard deviation (SD) unit increase in i.FCS, adjusting for key sociodemographic and lifestyle factors. Higher i.FCS was significantly associated with more favorable levels of nearly all risk factors evaluated (Table 2). For example, each SD increase in i.FCS (-10.9 points out of 100) was associated with lower BMI (−0.60 kg/m² [−0.70, −0.51]), systolic blood pressure (−0.69 mmHg [−0.91, −0.48]), LDL-C (−2.01 mg/dl [−2.63, −1.40]), HbA1c (−0.02% [−0.03, −0.01]); and fasting plasma glucose (−0.44 mg/dL [−0.74, −0.15]); and higher HDL-C (1.65 mg/d: [1.44, 1.85]). The association between i.FCS and triglyceride levels was nonsignificant (−1.55 mg/dL [−3.13, 0.03]). Further adjustment for total energy intake and NHANES cycle had no appreciable effect on the magnitude or statistical significance of these relationships (Table S7).

We used the same statistical methods to explore the association between an individual's FCS domain scores (separately derived from the product level domain score) and clinical risk factors. i.Nutrient

## Table 1 | Baseline characteristics of the national sample of 47,999 U.S. adults in this investigation, 1999–2018[a]

| Sociodemographic characteristics | n = 47,999 |
|---|---|
| *Age group, years* | |
| 20–34 | 12479 (27.9) |
| 35–49 | 11957 (28.2) |
| 50–64 | 11708 (25.5) |
| 65+ | 11855 (18.4) |
| Age, years | 47.2 (17.1) |
| *Sex* | |
| Male | 23029 (47.8) |
| Female | 24970 (52.2) |
| *Race/ethnicity* | |
| Mexican-American | 8419 (8.2) |
| Other Hispanic | 3908 (5.3) |
| Non-Hispanic White | 21590 (68.5) |
| Non-Hispanic Black | 9962 (11.1) |
| Asian/Other | 4120 (6.8) |
| *Education Level* | |
| <HS graduate | 12769 (17.0) |
| HS graduate | 11138 (24.1) |
| Some college or AA school | 13657 (31.2) |
| ≥College graduate | 10435 (27.8) |
| *Ratio of Family Income to Poverty[b]* | |
| <1.3 | 14734 (22.0) |
| 1.3–2.9 | 15384 (28.8) |
| 3+ | 17881 (49.2) |
| **Lifestyle and metabolic health characteristics** | |
| Alcohol use, percent total energy[c] | 2.7 (6.1) |
| Total physical activity, MET-hours/week[d] | 63.0 (103.7) |
| *Smoking status, n (%)* | |
| Non-smoker | 26067 (53.6) |
| Former smoker | 12039 (25.0) |
| Current smoker | 9893 (21.4) |
| Waist circumference, cm | 98.5 (16.4) |
| BMI, kg/m$^2$ | 28.8 (6.8) |
| HbA1c, % | 5.6 (0.9) |
| Fasting plasma glucose, mg/dL | 105.8 (30.8) |
| Total cholesterol, mg/dL | 196.7 (42.2) |
| HDL-cholesterol, mg/dL | 53.2 (16.3) |
| LDL-cholesterol, mg/dL | 119.0 (43.6) |
| Triglycerides, mg/dL | 122.7 (110.5) |
| Total cholesterol: HDL ratio | 4.0 (1.5) |
| Systolic blood pressure, mm Hg | 122.8 (18.0) |
| Diastolic blood pressure, mm Hg | 70.9 (12.5) |
| i.FCS[c] | 35.5 (10.9) |
| HEI 2015[c] | 57.3 (8.5) |
| Total energy intake, kcal/day[c] | 2090 (803) |
| *Energy reporting, TEI:BMR[e]* | |
| Underreport | 12062 (23.4) |
| Acceptable | 34623 (75.4) |
| Overreport | 574 (1.1) |
| **Presence of health conditions and medication usage** | |
| Metabolic syndrome[f] | 21935 (42.0) |
| Diabetes[g] | 8126 (12.9) |
| Cardiovascular disease[h] | 4713 (7.7) |
| Lung disease | 8709 (18.9) |

## Table 1 (continued) | Baseline characteristics of the national sample of 47,999 U.S. adults in this investigation, 1999–2018[a]

| Sociodemographic characteristics | n = 47,999 |
|---|---|
| Cancer | 4514 (9.8) |
| Optimal cardiometabolic health[i] | 2923 (7.4) |
| Diabetes medication | 5229 (8.0) |
| Cholesterol lowering medication | 8618 (16.3) |
| Hypertension medication | 14836 (26.3) |
| Angina medication | 645 (1.0) |

*AA* Associates Degree, *BMI* body mass index, *BMR* basal metabolic rate, *CI* confidence interval, *CVD* cardiovascular disease, *HbA1c* hemoglobin A1c, *HDL* high-density lipoprotein, *HEI* Healthy Eating Index, *HS* high school, *i.FCS* individual, dietary Food Compass Score, *LDL* low-density lipoprotein, *MET* metabolic equivalent of task, *SD* standard deviation, *TC:HDL* total cholesterol to HDL-cholesterol ratio, *TEI* total energy intake, *WC* waist circumference.

[a]Values are mean (SD) for continuous variables and N (%) for categorical variables. NHANES dietary recall sample weights were incorporated in all analyses to account for the complex survey design and response rates, providing nationally representative estimates for the non-institutionalized U.S. adult population. Any missing values were imputed using predictive mean matching, incorporating all available demographics, risk factors, and conditions data in the dataset. 81 fewer subjects were included in the prospective mortality analyses due to lack of linkage to the National Death Index.

[b]Represents the ratio of family income to the federal poverty threshold, adjusted for household size. A higher ratio indicates a higher level of income.

[c]For all dietary factors (i.e., percentage of total energy from alcohol (%); i.FCS; HEI 2015; and average total energy intake (in kcal/day): the value was calculated based on the total number of recall days available for each respondent. For example, when two days of data were available, an arithmetic average of that factor was calculated.

[d]Methodological differences in how physical activity was assessed in NHANES from 1999 to 2006 and from 2007 to 2018 preclude the use of a consistent variable for total physical activity across survey cycles. As such, we used a total physical activity variable (MET-hours/week) evaluated from 2007 to 2018, a leisure time physical activity (MET-hrs/week) variable for all survey cycles, and demographic and health data to impute total physical activity (MET-hrs/week) for 1999–2006 survey cycles using predictive mean matching. See Supplementary Materials (Supplementary Note 4, Table S13–15) for further methodological details.

[e]TEI:BMR, the ratio of average total energy intake (TEI, in kcal/day) to basal metabolic rate (BMR, kcal/day), was used to determine the plausibility of total energy reporting based on Goldberg cut-offs. BMR was estimated via the Schofield sex- and age-specific predictive equations based on weight and height. TEI was calculated based on the total number of recall days available for each respondent. For example, when two days of data were available, an arithmetic average TEI was calculated. Under-reporters, acceptable reporters and over-reporters were defined as having TEI:BMR < 0·96, 0·96–2·49 and >2·49 for 2-day dietary recall data, and < 0·87, 0·87–2·75 and >2·75 for 1-day dietary recall data, respectively, as previously derived in NHANES[57]. Over-reporters were excluded from sub-group analyses given sample size and number of events limitations.

[f]Defined according to AHA/NHLBI as the presence of three or more of: HDL-C < 40 mg/dL (men)/ <50 mg/dL (women) or on drug treatment for low HDL-C; triglycerides ≥150 mg/dL or on drug treatment for elevated triglycerides; blood pressure ≥130 SBP or ≥85 DBP or on antihypertensive drug treatment with a history of hypertension; WC ≥102 cm (men)/ ≥88 cm (women); and fasting plasma glucose ≥100 mg/dL or on drug treatment for elevated glucose[58].

[g]Defined based on presence of at least one of: HbA1c level > 6.5%, fasting plasma glucose > 126 mg/dL, or use of diabetes medications.

[h]Defined based on the presence of both (1) self-reported coronary heart disease, heart failure, stroke, myocardial infarction, or angina and also (2) cardiovascular medication use including use of angina, hypertension, or lipid medications.

[i]Defined based on optimal levels for each of the following: adiposity (BMI < 25 kg/m$^2$ and WC ≤ 88 cm (women)/WC ≤ 102 cm (men)); blood glucose (FPG < 100 mg/dL and HbA1c < 5.7% and not taking diabetes medication); blood lipids (TC:HDL < 3.5:1 and not taking lipid lowering medication); blood pressure (SBP < 120 mmHg, DBP < 80 mmHg and not taking blood-pressure lowering medication); and history of CVD (no self-reported CHD, heart failure, myocardial infarction, stroke, or angina).

Ratios, i.Food Ingredients, and i.Minerals each generally showed stronger protective associations with clinical risk factors, while i.Fiber and Protein, i.Processing, and i.Phytochemicals generally showed less robust associations (Table 3). Considering each major clinical risk factor, i.Nutrient Ratios had the strongest protective association with BMI (−0.47 kg/m$^2$ [−0.56, −0.39]), diastolic blood pressure (−0.52 mmHg [−0.69, −0.36]), LDL cholesterol (−2.12 mg/dL [−2.76, −1.47]), HDL-cholesterol (1.50 mg/dL [1.29, 1.71]), and the TC:HDL ratio (−0.14 [−0.16, −0.12]); and i.Food Ingredients with systolic blood pressure (−0.86 mmHg [−1.07, −0.64]), triglycerides (−3.23 mg/dL [−4.73, −1.73]), HbA1c, (−0.01% [−0.02, −0.01]), and fasting plasma glucose (−0.44 mg/dL [−0.73, −0.14]). However, an individual's FCS

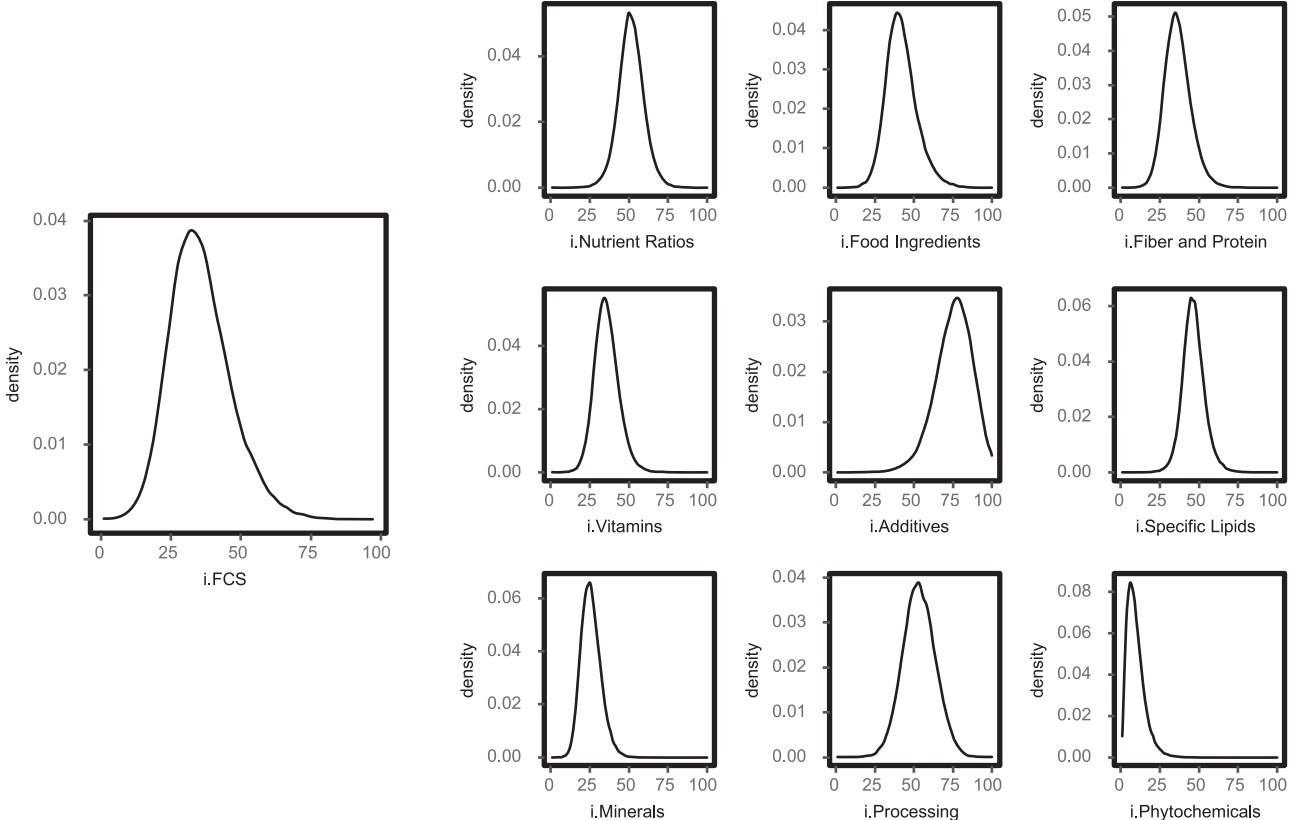

**Fig. 1 | Population energy-weighted, individual Food Compass Score (i.FCS) and individual component Domain Score densities among U.S. adults, 1999–2018.** Density plots reflect the population distribution of i.FCS among 47999 U.S. adults from 1999 to 2018. i.FCS was calculated as the energy-weighted mean of the Food Compass Score of all foods consumed, as reported in all available 24 hr recalls, ranging from 1 to 100. The same methods were used for calculating the component domain scores at the individual level. i.FCS individual Food Compass Score.

domain score generally had significant protective associations with at least several of the different risk factors.

**Validation of Food Compass against prevalent health conditions**
To assess how a person's i.FCS was cross-sectionally associated with prevalent health conditions in NHANES (metabolic syndrome, cardiovascular disease, diabetes, lung disease, and optimal cardiometabolic health [optimal levels of a constellation of 5 cardiometabolic risk factors; see table S3 for detailed definition]), we evaluated survey-weighted logistic regression models for each standard deviation (SD) unit increase in i.FCS, adjusting for key sociodemographic and lifestyle factors. Higher levels of i.FCS were significantly associated with lower risk of several prevalent health conditions (Table 2). For example, each SD increase in i.FCS (-10.9 points) was associated with 15% lower prevalence of metabolic syndrome (OR = 0.85 [95%CI: 0.82, 0.88]), 8% lower prevalence of CVD (0.92 [0.88, 0.96]), 5% lower prevalence of cancer (0.95 [0.91, 0.99]),and 8% lower prevalence of lung disease (0.92 [0.88, 0.96]). A nonsignificant trend was seen toward lower prevalence of diabetes (0.96 [0.90, 1.01]). Each SD increase in i.FCS was also associated with 24% higher prevalence of optimal cardiometabolic health (1.24 [1.16, 1.32]). Further adjustment for total energy intake and NHANES cycle had no appreciable effects on the magnitude or statistical significance of results (Table S7).

We used the same statistical methods to explore the association between individual FCS domains (separately derived from the product level domain score) and prevalent health conditions. Each individual FCS domains was associated with a lower risk of metabolic syndrome, with i.Nutrient Ratios having the strongest inverse association (0.86 [0.84, 0.89]) (Table 3). For diabetes, i.Nutrient Ratios had the strongest inverse association (0.94 [0.89, 0.99]); for CVD, i.Minerals, i.Food

Ingredients, i.Additives, i.Processing, i.Fiber and Protein, and i.Specific Lipids were each inversely associated (OR 0.91 to 0.95 each); for cancer, i.Nutrient Ratios, i.Additives, i.Processing and i.Fiber and Protein were each inversely associated (OR 0.93 to 0.95 each); and for lung disease, all domains except i.Food Ingredients were inversely associated. Seven of 9 individual domain scores (except i.Vitamins and i.Fiber and Protein) were also associated with higher prevalence of optimal cardiometabolic health. In further post-hoc exploratory analyses disaggregating the i.Fiber and Protein domain into its individual attributes (i.Fiber and i.Protein), i.Fiber was marginally associated with lower prevalence of diabetes (0.97 [0.92, 1.03]), while i.Protein was associated with higher prevalence of diabetes (1.10 [1.04, 1.16]).

**Validation of Food Compass against all-cause and cause-specific mortality**
To assess the prospective association between a person's i.FCS and total mortality, we used linked, cause-specific mortality data from the National Death Index through 2018. Survey-weighted Cox proportional hazard models were used to estimate the hazard ratio (HR) and 95% CI between a one SD in i.FCS and risk of all-cause and cause-specific (cardiometabolic and cancer) mortality, adjusting for key sociodemographic and lifestyle factors, and with study time as the time variable. Over a total of 20.8 years of follow-up (with participants contributed person-time from the time of their enrollment until death or end-of-follow-up on January 1, 2019, whichever came first), 7481 deaths occurred, including 2619 (35.0%) due to cardiometabolic disease and 1691 (22.6%) due to cancer. In multivariable-adjusted analyses, each 1 SD (10.9) increase in i.FCS was prospectively associated with a 7% lower risk of all-cause mortality (HR = 0.93 [0.89, 0.96]) (Fig. 4, Table S8). For comparison, similar

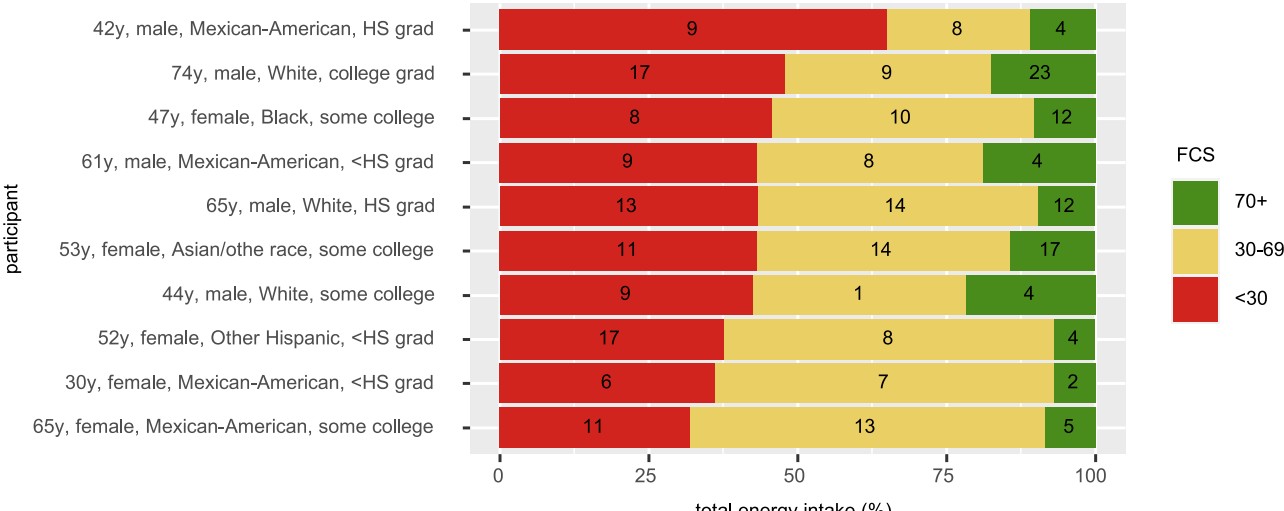

**Fig. 2 | Examples of individuals having the mean U.S. i.FCS of 35.5, demonstrating variation in the consumption of products with FCS ≤ 30, 31–69, and ≥70 that achieves a similar overall score.** The distribution (count and contribution to total energy intake [%]) of consumed food and beverage products, based on previously defined product healthfulness thresholds (i.e., FCS ≤ 30 as products to minimize; FCS 31-69 as products to be consumed in moderation; and FCS ≥ 70 as products to be encouraged) was assessed for all NHANES participants. Examples are shown for 10 participants with i.FCS at the U.S. mean (35.5 ±1) and consuming at least 10 different products over 2 days of 24 hr recalls. The numbers within each stacked bar graph indicate the count of food and beverage products consumed from that category of FCS across two days of reported intake; and the color bars represent the percentage energy contribution of food and beverage products from that category of FCS across two days of reported intake. FCS Food Compass Score, grad graduate, HS high school, i.FCS individual Food Compass Score.

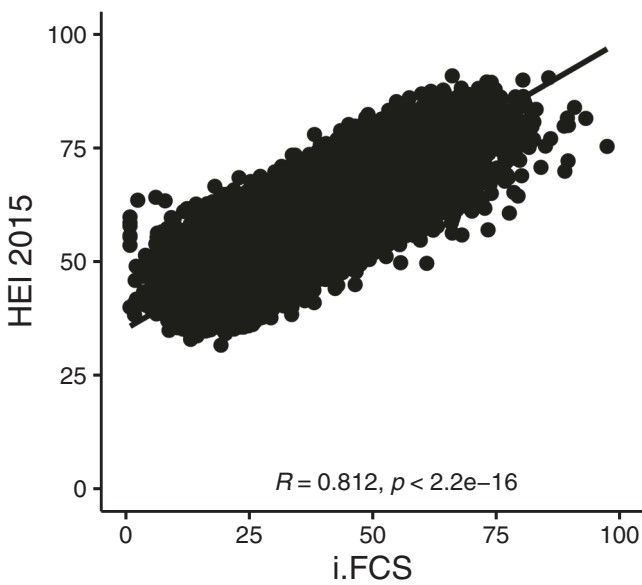

**Fig. 3 | Relationship between the Healthy Eating Index (HEI) 2015 and energy-weighted, individual Food Compass Score (i.FCS) among U.S. adults, 1999–2018.** i.FCS was calculated by taking the energy-weighted mean of the respective score for all foods consumed by that individual reported in up to two 24 hr recalls. Black dots represent each NHANES respondent; solid line, the line of best fit between all NHANES respondents; and R value, the correlation and p-value (2-sided) for the Spearman correlation. i.FCS individual Food Compass Score, NHANES National Health and Nutrition Examination Survey.

lower risk of all-cause mortality was seen among individuals with some college education compared to individuals with less than high school education (0.94 [0.86, 1.02] and for a 1 SD increase in total physical activity (0.90 [0.85, 0.94]). i.FCS was associated with a trend toward lower cancer-specific mortality (0.92 [0.85, 1.00]); and not significantly associated with cardiometabolic mortality (0.95 [0.89, 1.02]).

Further adjustment for clinical risk factors, prevalent conditions, and medications, which could each be confounders or mediators, did not appreciably alter these findings (Table S8). Censoring deaths at 85+ years as well as further adjustment for energy intake and survey cycle, slightly strengthened the magnitude of the protective association between i.FCS and all-cause mortality (0.91 [0.87, 0.95] and cancer mortality (0.92 [0.89, 0.96]) (Table S9). In addition, when censoring deaths at 85+ years, the association of i.FCS with CMD-specific mortality became marginally significant (0.92 [0.85, 1.00]).

In exploratory analyses, we investigated whether the relationship between i.FCS and total mortality varied in population subgroups according to age, sex, race/ethnicity, education, income, BMI, and plausibility of energy reporting. Findings were similar across subgroups, with no significant differences in the observed protective associations between i.FCS and mortality (p-interaction >0.05 each) (Table S10).

We used the same statistical methods to explore the association between individual FCS domains (separately derived from the product level domain score) and mortality. Exploring individual FCS domains, significant inverse associations with all-cause mortality were seen for i.Nutrient Ratios (0.92 [0.89, 0.96]), i.Food Ingredients (0.95 [0.91, 0.99]), i.Additives (0.95 [0.92, 0.98]), i.Processing (0.93 [0.89, 0.96]), i.Fiber and Protein (0.93 [0.90, 0.96]), and i.Specific Lipids (0.96 [0.92, 1.00]) (Table S8).

Using semi-parametric, restricted cubic splines, the relationship between i.FCS and all-cause mortality appeared potentially nonlinear, with a stronger protective association until an i.FCS of ~40 (approximately the 75th percentile score), with a less strong inverse relationship thereafter, but this potential nonlinearity was not statistically significant (p-nonlinearity = 0.12) (Fig. S3).

## Discussion

Based on nationally representative data among U.S. adults, we assessed individuals' dietary habits based on the Food Compass-derived nutrient profiling scores of the individual food and beverage products they consumed. An individual's dietary quality assessed by Food Compass (i.FCS) was highly correlated with HEI-2015 (R = 0.81),

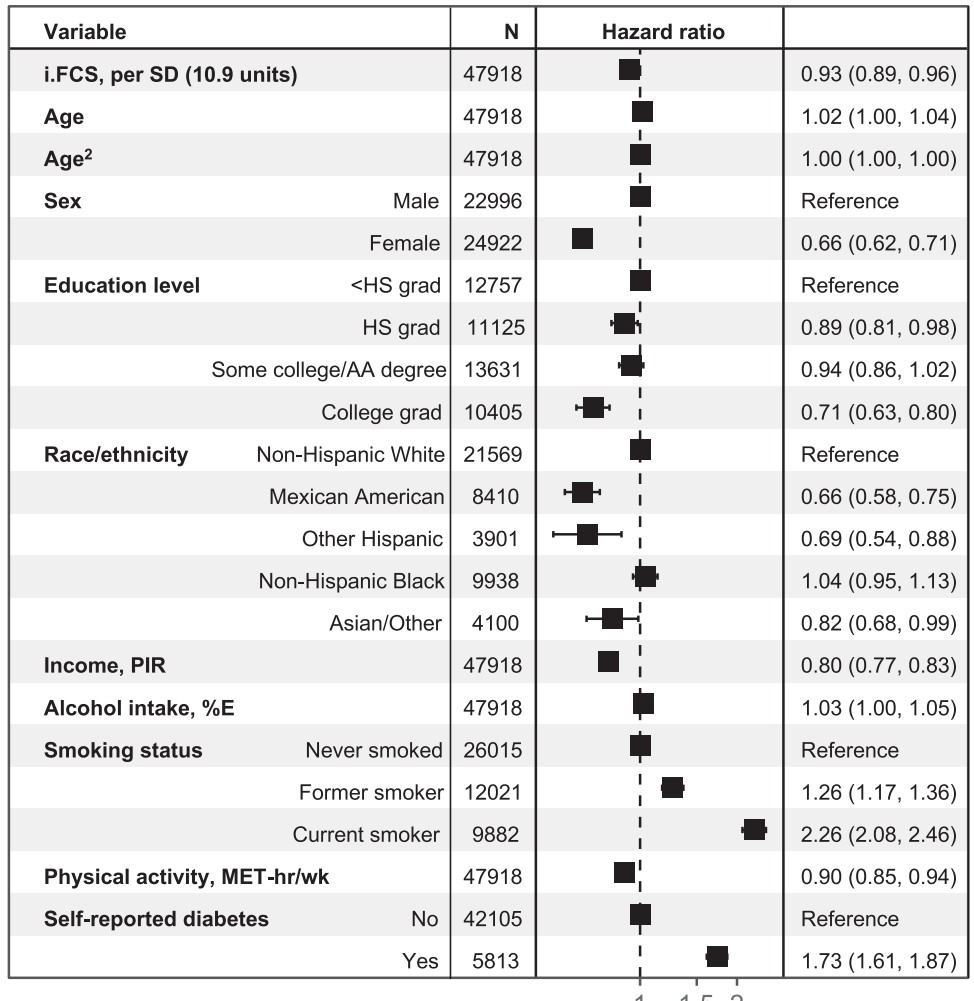

| Variable | | N | Hazard ratio | |
|---|---|---|---|---|
| **i.FCS, per SD (10.9 units)** | | 47918 | | 0.93 (0.89, 0.96) |
| **Age** | | 47918 | | 1.02 (1.00, 1.04) |
| **Age²** | | 47918 | | 1.00 (1.00, 1.00) |
| **Sex** | Male | 22996 | | Reference |
| | Female | 24922 | | 0.66 (0.62, 0.71) |
| **Education level** | <HS grad | 12757 | | Reference |
| | HS grad | 11125 | | 0.89 (0.81, 0.98) |
| | Some college/AA degree | 13631 | | 0.94 (0.86, 1.02) |
| | College grad | 10405 | | 0.71 (0.63, 0.80) |
| **Race/ethnicity** | Non-Hispanic White | 21569 | | Reference |
| | Mexican American | 8410 | | 0.66 (0.58, 0.75) |
| | Other Hispanic | 3901 | | 0.69 (0.54, 0.88) |
| | Non-Hispanic Black | 9938 | | 1.04 (0.95, 1.13) |
| | Asian/Other | 4100 | | 0.82 (0.68, 0.99) |
| **Income, PIR** | | 47918 | | 0.80 (0.77, 0.83) |
| **Alcohol intake, %E** | | 47918 | | 1.03 (1.00, 1.05) |
| **Smoking status** | Never smoked | 26015 | | Reference |
| | Former smoker | 12021 | | 1.26 (1.17, 1.36) |
| | Current smoker | 9882 | | 2.26 (2.08, 2.46) |
| **Physical activity, MET-hr/wk** | | 47918 | | 0.90 (0.85, 0.94) |
| **Self-reported diabetes** | No | 42105 | | Reference |
| | Yes | 5813 | | 1.73 (1.61, 1.87) |

1    1.5  2

**Fig. 4 | Prospective associations of the individual Food Compass Score (i.FCS), sociodemographic factors, and other factors with all-cause mortality among U.S. adults, 1999–2018.** The i.FCS for each person was calculated as the energy-weighted mean of the FCS of all unique foods and beverages consumed, based on up to two 24 hr recalls per person, with a potential range from 1 to 100. Boxes represent the multivariable-adjusted hazard ratio; and error bars, the 95% CI; based on survey-weighted Cox proportional hazard models incorporating NHANES dietary recall sample weights to account for the complex survey design and response rates and provide nationally representative estimates for the non-institutionalized U.S. population. All variables in the Figure were included together in the multivariable model. AA Associates Degree, grad graduate, HS high school, i.FCS individual Food Compass Score, MET metabolic equivalent of task, %E percent energy, PIR ratio of family income to poverty threshold, SD standard deviation.

validating the use of Food Compass scoring of specific food and beverage products for adequately reflecting dietary quality. Furthermore, after adjustment for other major risk factors, i.FCS was associated with more favorable levels of major clinical risk factors, major disease conditions, and optimal cardiometabolic health. Prospectively, each SD increase in i.FCS (~11 points out of 100) was associated with a 7% lower risk of all-cause mortality. Taken together, these findings support the validity of the FCS as a tool to score the healthfulness of individual food and beverage products.

These findings support the original design intent of Food Compass – to incorporate the best available science on health impacts of nutrients, food ingredients, and other emerging food and beverage characteristics of public health concern, selected and weighted to identify more healthful products[15]. Notably, different people could arrive at a similar i.FCS with very different combinations of healthier or less healthy foods, and yet the overall i.FCS still associated with a healthy diet pattern, health risk factors, prevalent disease conditions, and total mortality. Together with our previous demonstration of content validity, discriminatory validity, and convergent validity of the Food Compass when applied to individual products[15], the current demonstration of construct validity against population diet quality,

risk factors, health conditions, and mortality lends further support to this NPS as a valid measure of product healthfulness.

The identified associations for individual domains also support biologic plausibility of the findings. For example, the i.Nutrient Ratios domain was associated with blood lipid levels, adiposity, blood pressure, blood glucose levels, prevalent metabolic syndrome, and prevalent diabetes supported by robust evidence for effects of dietary fat quality on these outcomes[20], of carbohydrate quality (i.e., total carbohydrate: fiber ratio) on adiposity and diabetes risk[21,22], and of the joint intakes of potassium and sodium on blood pressure and stroke risk[23]. The i.Food Ingredients domain was also robustly associated with several major health endpoints, consistent with the established etiologic associations of the included healthful (fruits, non-starchy vegetables, beans/legumes, whole grains, nuts/seeds, seafood, yogurt, plant oils) and unhealthful (refined grains, red/processed meat) attributes in this domain[24,25]. i.Processing was associated with BMI, consistent with interventional evidence suggesting that diets high in ultra-processed foods cause excess caloric intake and weight gain[26]. The less robust (or nonsignificant) associations of i.Phytochemicals and i.Specific Lipids with several health endpoints are consistent with the more emerging nature of scientific evidence for causal effects of flavonoids,

**Table 2 | Multivariable-adjusted, cross-sectional associations of the individual Food Compass Score (i.FCS) with major clinical risk factors and prevalent health conditions among U.S. adults, 1999–2018**

| | i.FCS, per SD (10.9 points)[a] |
|---|---|
| **Clinical risk factors** | **Difference (95% CI)[b]** |
| BMI, kg/m$^2$ | −0.60 (−0.70, −0.51) |
| Systolic blood pressure, mmHg | −0.69 (−0.91, −0.48) |
| Diastolic blood pressure, mmHg | −0.49 (−0.66, −0.32) |
| LDL-C, mg/dL | −2.01 (−2.63, −1.40) |
| HDL-C, mg/dL | 1.65 (1.44, 1.85) |
| Triglycerides, mg/d | −1.55 (−3.13, 0.03) |
| TC: HDL ratio, 1 unit | −0.13 (−0.15, −0.12) |
| HbA1c, % | −0.02 (−0.03, −0.01) |
| Fasting plasma glucose, (mg/dL) | −0.44 (−0.74, −0.15) |
| **Prevalent health conditions** | **Odds ratio (95% CI)[b]** |
| Metabolic syndrome[c] | 0.85 (0.82, 0.88) |
| Diabetes[d] | 0.96 (0.91, 1.01) |
| Cardiovascular disease[e] | 0.92 (0.88, 0.96) |
| Cancer | 0.95 (0.91, 0.99) |
| Lung disease | 0.92 (0.88, 0.96) |
| Optimal cardiometabolic health[f] | 1.24 (1.16, 1.32) |

*BMI* body mass index, *CI* confidence interval, *CVD* cardiovascular disease, *HbA1c* hemoglobin A1c, *HDL-C* high-density lipoprotein cholesterol, *i.FCS* individual, dietary Food Compass Score, *LDL* low density lipoprotein cholesterol, *TC:HDL* total cholesterol to HDL-cholesterol ratio.

[a]Individual Food Compass Score (i.FCS) calculated as the energy-weighted mean of FCS of all foods consumed, as reported in up to two 24 hr recalls per person, with a potential range from 1 to 100.

[b]Survey-weighted, multivariable-adjusted linear or logistic regression models incorporated NHANES dietary recall sample weights to account for the complex survey design and response rates and provide nationally representative effect estimates and associated 95% confidence interval for the non-institutionalized U.S. population. All models adjusted for: age (years), age$^2$ (years), sex, race/ethnicity (Mexican-American, other Hispanic, non-Hispanic White, non-Hispanic Black, Asian/other race), education level (<HS graduate, HS graduate, some college or associates degree, ≥ college graduate), income (poverty: income ratio), smoking status (non-smoker, former smoker, current smoker); total physical activity (MET-hours/week), alcohol use (%energy), and self-reported diabetes (yes, no).

[c]Defined according to AHA/NHLBI as the presence of three or more of: HDL-C < 40 mg/dL (men)/ <50 mg/dL (women) or on drug treatment for low HDL-C; triglycerides ≥150 mg/dL or on drug treatment for elevated triglycerides; blood pressure ≥130 SBP or ≥85 DBP or on antihypertensive drug treatment with a history of hypertension; WC ≥102 cm (men) ≥88 cm (women); and fasting plasma glucose ≥100 mg/dL or on drug treatment for elevated glucose[58].

[d]Defined based on presence of at least one of: HbA1c level > 6.5%, fasting plasma glucose > 126 mg/dL, or use of diabetes medications.

[e]Defined based on the presence of both (1) self-reported coronary heart disease, heart failure, stroke, myocardial infarction, or angina and also (2) cardiovascular medication use including use of angina, hypertension, or lipid medications.

[f]Defined based on optimal levels for each of the following: adiposity (BMI < 25 kg/m$^2$ and WC ≤ 88 cm (women)/WC ≤ 102 cm (men)); blood glucose (FPG < 100 mg/dL and HbA1c < 5.7% and not taking diabetes medication); blood lipids (TC:HDL < 3.5:1 and not taking lipid lowering medication); blood pressure (SBP < 120 mmHg, DBP < 80 mmHg and not taking blood-pressure lowering medication); and history of CVD (no self-reported CHD, heart failure, myocardial infarction, stroke, or angina).

carotenoids, and specific lipids on major health outcomes, and support the lower weighting of these domains in the Food Compass scoring[15]. The observed harmful association between the i.Protein attribute and prevalent diabetes is consistent with meta-analyses of prospective cohort studies which identified a positive association between higher protein intake and higher incidence diabetes[27]; as well as randomized trials where higher protein intake was associated with hepatic de novo lipogenesis[28]—a driving pathway for fatty liver, visceral fat accumulation, and insulin resistance.

One striking finding was the low average i.FCS across the U.S. adult population. The mean score (-36) was not much higher than the identified threshold score (FCS ≤ 30) for individual food and beverage products to be minimized in the diet[15], and fewer than 1% of U.S. adults had an i.FCS above 70 (the identified threshold score for food and beverage

products to be encouraged). These results suggest that, based on NPS scoring of individual foods and beverages, the leading caloric contributors to American's diets are, on average, mostly foods and beverages to be minimized or avoided. Using the cut-points of ≤ 30, 31–69, and ≥ 70 for i.FCS, we identified 32.7% of U.S. adults having poor diet quality, 66.8%, intermediate; and 0.5%, ideal. These results using an NPS of individual foods and beverages are consistent with previous studies based on validated diet pattern scores, which reported 46% of U.S. adults having poor diet quality; 53%, intermediate diet quality; and only 1%, ideal diet quality[29,30]. Our findings further highlight the suboptimal state of U.S. nutrition, and the urgent need for multistakeholder policy, business, and other systems interventions to improve diet quality.

In the U.S., the FDA is currently finalizing a definition to permit industry use of the word "healthy" on food and beverage products, including plans for a corresponding "FDA healthy" front-of-package icon[31]. Separately, major food retailers are also designing their own, different, industry-developed nutrient profiling systems for product labeling on their shelves and online[7–9]. In the European Union, seven countries have pledged to support coordination for promoting a unified front-of-pack (FOP) label[5], with further calls by non-governmental bodies to adapt nutrient profiling as a tool to guide consumer choice[32]. The Food Compass could be a possible unified standard NPS for these efforts, given its association with a healthy diet pattern and multiple health endpoints and its use of a single, consistent algorithm for all foods, beverages, mixed ingredients, and mixed meals. Such a standard may also provide a more accurate assessment of healthfulness for consumers, industry, and procurement decisions, in comparison to strategies using isolated nutrients or ingredients, such as the FOP warning labels recently implemented in Chile, Mexico, Uruguay, and Israel[33]. While the Food Compass has more attributes than existing NPS, its core design strengths are the use of domains, which provides a holistic assessment of foods and beverages while also preventing the excess weight from any single attribute; integration of cutting edge science in selection of attributes (i.e., nutrient ratios which more accurately capture fat, mineral and carbohydrate quality) and omission of outdated attributes such as total fat or total calories; scoring on a basis of 100 kcal rather than 100 g, which greatly reduces bias from water weight; and incorporation of novel relevant features such as processing and additives. In future work, we plan to test FCS versions that retain these core design strengths while employing a more limited set of attributes; as well as leverage multivariable imputation to handle missing values in certain attributes in large datasets of branded products. Extension to datasets in low and middle income countries will also be important.

While FOP labeling of retail products is timely and relevant, an NPS like Food Compass that scores mixed meals and complex foods has broader applications. In this national dataset, Food Compass was able to score not only manufactured products but a person's entire diet, including complex home-cooked, cafeteria, and restaurant mixed meals which other major NPS generally cannot do. This raises the potential for nutrient profiling for labeling of restaurant menus; as well as for expanding economic incentives and subsidies beyond fruits and vegetables in programs like the U.S. Supplemental Nutrition Assistance Program (SNAP)[34] and insurance wellness programs[35]. Similarly, food manufacturer pledges for product reformulation, often based on internally selected goals for isolated single nutrients[36,37], could be based on an independently derived, validated NPS. And, socially-driven investment metrics can incorporate nutrition and health by leveraging NPS to identify companies with healthier product portfolios and sales[4,12]. NPS like Food Compass can be one important tool in such strategies, allowing consumers, industry (including food manufacturers, retailers, and restaurants), investors, schools, hospitals, worksites, and policy makers to identify and shift toward healthier food and beverage options. Our current findings also support flexibility for these different stakeholders in improving healthfulness of their portfolios, as different people arrived at their i.FCS scores with

**Table 3 | Multivariable adjusted, survey-weighted, cross-sectional analysis of the association between the 9 components of the Food Compass Score at the individual level (i.Domain Scores) and cardiometabolic health among U.S. adults, 1999–2018**

| | i.Domain Scores[a] | | | | | | | | |
|---|---|---|---|---|---|---|---|---|---|
| | i.Nutrient Ratios[b] | i.Vitamins | i.Minerals | i.Food Ingredients[c] | i.Additives[c] | i.Processing | i.Fiber and Protein[c] | i.Specific Lipids | i.Phytochemicals |
| *Continuous biomarkers (difference [95% CI])[a]* | | | | | | | | | |
| BMI, kg/m² | −0.47 (−0.56, −0.39)*** | −0.27 (−0.37, −0.18)*** | −0.46 (−0.55, −0.37)*** | −0.46 (−0.54, −0.37)*** | −0.21 (−0.31, −0.1)*** | −0.30 (−0.38, −0.21)*** | −0.04 (−0.13, 0.06) | −0.16 (−0.26, −0.07)*** | −0.32 (−0.41, −0.23)*** |
| Systolic blood pressure, mmHg | −0.62 (−0.85, −0.40)*** | −0.27 (−0.5, −0.05)* | −0.50 (−0.7, −0.30)*** | −0.86 (−1.07, −0.64)*** | 0.16 (−0.05, 0.37) | −0.04 (−0.25, 18) | −0.06 (−0.28, 16) | −0.24 (−0.47, −0.01)* | −0.30 (−0.49, −0.12)** |
| Diastolic blood pressure, mmHg | −0.52 (−0.69, −0.36)*** | −0.35 (−0.49, −0.2)*** | −0.42 (−0.57, −0.27)*** | −0.35 (−0.51, −0.19)*** | −0.18 (−0.34, −0.02)* | −0.08 (−0.23, 0.08) | −0.34 (−0.5, −0.18)*** | 0.01 (−0.14, 0.16) | −0.26 (−0.42, −0.11)** |
| LDL-C, mg/dL | −2.12 (−2.76, −1.47)*** | −1.24 (−1.81, −0.66)*** | −0.96 (−1.51, −0.42)*** | −1.69 (−2.24, −1.15)*** | −0.88 (−1.46, −0.30)** | −0.32 (−0.97, 0.33) | −0.88 (−1.47, −0.29)** | −1.46 (−2.07, −0.85)*** | −0.94 (−1.53, −0.34)** |
| HDL-C, mg/dL | 1.50 (1.29, 1.71)*** | 0.80 (0.61, 0.98)*** | 0.98 (0.8, 1.16)*** | 1.29 (1.09, 1.50)*** | 1.19 (0.99, 1.39)*** | 0.79 (0.58, 1.00)*** | 1.10 (0.92, 1.29)*** | 0.67 (0.47, 0.86)*** | 0.54 (0.35, 0.72)*** |
| Triglyceride, mg/dL | −1.87 (−3.49, −0.25)* | 0.03 (−1.35, 1.41) | 0.50 (−1.02, 2.01) | −3.23 (−4.73, −1.73)*** | 0.50 (−0.88, 1.88) | 1.20 (−0.45, 2.84) | 0.36 (−1.12, 1.84) | −2.35 (−3.89, −0.8)** | −0.60 (−2.08, 0.88) |
| TC: HDL ratio, 1 unit | −0.14 (−0.16, −0.12)*** | −0.07 (−0.09, −0.05)*** | −0.07 (−0.09, −0.05)*** | −0.12 (−0.13, −0.10)*** | −0.08 (−0.1, −0.06)*** | −0.04 (−0.06, −0.02)*** | −0.07 (−0.09, −0.05)*** | −0.08 (−0.1, −0.05)*** | −0.05 (−0.07, −0.04)*** |
| HbA1c, % | −0.01 (−0.02, 0.00)* | −0.01 (−0.02, 0.00)* | −0.01 (−0.02, −0.01)** | −0.01 (−0.02, −0.01)** | 0.01 (0.0, 0.01) | −0.01 (−0.02, 0.0)** | 0.00 (−0.01, 0.01) | 0.00 (−0.01, 0.01) | −0.01 (−0.02, −0.01)** |
| Fasting plasma glucose, mg/dL | −0.32 (−0.63, −0.01)* | −0.28 (−0.58, 0.01) | −0.35 (−0.65, −0.06)* | −0.44 (−0.73, −0.14)** | 0.33 (0.07, 0.6)* | −0.17 (−0.48, 0.15) | 0.04 (−0.26, 0.34) | −0.15 (−0.45, 0.15) | −0.22 (−0.51, 0.08) |
| *Prevalent health conditions (odds ratio [95% CI])[a]* | | | | | | | | | |
| Metabolic syndrome[d] | 0.86 (0.84, 0.89)*** | 0.94 (0.91, 0.97)*** | 0.90 (0.87, 0.93)*** | 0.88 (0.85, 0.91)*** | 0.93 (0.9, 0.96)*** | 0.94 (0.91, 0.97)*** | 0.97 (0.93, 1.00)* | 0.96 (0.93, 0.99)* | 0.92 (0.89, 0.95)*** |
| Diabetes[e] | 0.94 (0.89, 0.99)* | 1.03 (0.97, 1.09) | 0.99 (0.93, 1.05) | 0.94 (0.90, 1.00)* | 1.04 (0.99, 1.1) | 1.00 (0.95, 1.05) | 1.07 (1.01, 1.13)* | 0.96 (0.91, 1.02) | 0.97 (0.91, 1.03) |
| Cardiovascular disease[f] | 0.96 (0.91, 1.01) | 1.00 (0.95, 1.04) | 0.95 (0.90, 0.99)* | 0.94 (0.89, 0.99)* | 0.92 (0.88, 0.97)*** | 0.91 (0.87, 0.96)*** | 0.94 (0.89, 0.99)* | 0.95 (0.90, 1.00)* | 1.01 (0.97, 1.05) |
| Cancer | 0.94 (0.90, 0.99)* | 0.96 (0.91, 1.00) | 0.96 (0.92, 1.01) | 1.02 (0.97, 1.06) | 0.93 (0.89, 0.97)** | 0.95 (0.90, 1.00)* | 0.93 (0.88, 0.98)** | 1.02 (0.98, 1.07) | 1.00 (0.94, 1.05) |
| Lung disease | 0.90 (0.9, 0.97)*** | 0.93 (0.90, 0.96)*** | 0.91 (0.87, 0.95)*** | 1.00 (0.96, 1.04) | 0.93 (0.89, 0.96)*** | 0.92 (0.89, 0.96)*** | 0.90 (0.87, 0.93)*** | 1.05 (1.01, 1.08)** | 0.95 (0.92, 0.99)* |
| Optimal cardiometabolic health[g] | 1.22 (1.15, 1.29)*** | 1.05 (0.99, 1.11) | 1.11 (1.05, 1.18)*** | 1.20 (1.13, 1.29)*** | 1.10 (1.03, 1.17)** | 1.09 (1.03, 1.15)** | 1.03 (0.98, 1.14) | 1.08 (1.03, 1.14)** | 1.12 (1.07, 1.17)*** |

*BMI* body mass index, *CI* confidence interval, *CVD* cardiovascular disease, *HbA1c* hemoglobin A1c, *HDL-C* high-density lipoprotein cholesterol, *i.FCS* individual, dietary Food Compass Score, *LDL-C* low density lipoprotein cholesterol, *TC:HDL* total cholesterol to HDL-cholesterol ratio.

*P < 0.05, **P < 0.01, ***P < 0.001.

a i.Domain Scores represent the nine component domains of the Food Compass Score at the individual level. For each i.Domain Score, the energy-weighted mean for all reported foods consumed, as reported in 24 hr dietary recalls, was calculated, as with i.FCS. Differences and odds ratios reflect a 1 standard deviation increase in respective i.Domain Score.

b Survey-weighted, multivariable-adjusted linear regression (or logistic regression) models incorporated NHANES dietary sample weights to account for the complex survey design and response rates and provide nationally representative effect estimates (or odds ratios) and associated 95% confidence interval for the non-institutionalized U.S. population. All models adjusted for: age (years), sex, race/ethnicity (Mexican-American, other Hispanic, non-Hispanic White, non-Hispanic Black, Asian/other race), education level (<HS graduate, HS graduate, some college or associates degree, ≥ college graduate), income (poverty: income ratio), smoking status (non-smoker, former smoker, current smoker); total physical activity (MET-hours/week), alcohol use (%energy), and self-reported diabetes (yes, no).

c The i.Additives domain, of which added sugar is a major component, was cross-sectionally associated with more favorable levels of BMI, diastolic blood pressure, LDL-C, HDL-C, TC:HDL, metabolic syndrome, CVD, cancer, lung disease, and optimal cardiometabolic health, but also higher fasting glucose and nonsignificant trend toward higher prevalence of diabetes; suggesting potential for some residual reverse causation (people with higher blood sugar avoiding added sugar in their diets). Similarly, the i.Fiber and Protein domain was cross-sectionally associated with more favorable levels of diastolic blood pressure, LDL-C, HDL-C, TC:HDL, metabolic syndrome, CVD, cancer, and lung disease, but also higher prevalence of diabetes, which was driven by the protein component when attributes were separately evaluated. This could relate to reverse causation, adverse effects of excess protein on hepatic de novo lipogenesis, or other adverse compounds, like heme iron, in high protein foods.

d Defined according to AHA/NHLBI as the presence of three or more of: HDL-C <40 mg/dL (men)/<50 mg/dL (women) or on drug treatment for low HDL-C; triglycerides ≥150 mg/dL or on drug treatment for elevated triglycerides; blood pressure ≥130 SBP or ≥85 DBP or on antihypertensive drug treatment with a history of hypertension; WC ≥102 cm (men)/ ≥88 cm (women); and fasting plasma glucose ≥100 mg/dL or on drug treatment for elevated glucose[58].

e Defined based on presence of at least one of: HbA1c level > 6.5%, fasting plasma glucose > 126 mg/dL, or use of diabetes medications.

f Defined based on the presence of both (1) self-reported coronary heart disease, heart failure, stroke, myocardial infarction, or angina and also (2) cardiovascular medication use including use of angina, hypertension, or lipid medications.

g Defined based on optimal levels for each of the following: adiposity (BMI ≥ 25 kg/m² and WC ≤ 88 cm (women)/WC ≤ 102 cm (men)): blood glucose (FPG < 100 mg/dL and HbA1c < 5.7% and not taking diabetes medication): blood lipids (TC:HDL < 3.5:1 and not taking lipid lowering medication): blood pressure (SBP < 120 mmHg, DBP <80 mmHg and not taking blood-pressure lowering medication); and history of CVD (no self-reported CHD, heart failure, myocardial infarction, stroke, or angina).

very different combinations of healthier or less healthy foods, and yet still the overall i.FCS was predictive of HEI, health risk factors, prevalent disease conditions, and total mortality.

Further analyses are required to assess Food Compass's comparative utility to other NPS across these various public and private use cases. Additionally, we have not compared the validity of different NPS against health outcomes, and in different nations and subpopulations – an important area for future work. Finally, the long-term vision of Food Compass is to score additional features of foods and beverages, such as environmental sustainability, social justice, and animal welfare – one for each direction of the compass. Future work is required to explore, add, and validate these additional dimensions.

The ability to score food and beverage items across 9 component domains is an important feature of the FCS and current investigation. Among the 9 domains, applied to individual items and then energy-weighted to persons' diets, none by itself had as strong an association with HEI, nor with the full range of health risk factors, prevalent diseases, and total mortality, as the overall i.FCS. At the same time, each i.Domain Score had some associations with HEI as well as varying strengths of associations with different health endpoints. These findings support the complementary nature of the different domains, each providing supportive and somewhat distinct information, as components of the Food Compass. The growing evidence for complex, heterogenous effects of diet on health (e.g., via the gut microbiota, epigenetics, etc.)[38] further supports the utility of a more holistic, multi-domain measure of healthfulness of foods and beverages. In addition, the domain structure ensures that no one domain or attributes can drive the overall product score—a mechanism which prevents mis-scoring based on extreme values of a single or few nutrients as well as industry "gaming" the system by fortifying food products with isolated vitamins. This holistic, domain-based scoring also permits a single, consistent scoring algorithm across all food and beverage products, including mixed meals – in contrast to all other major current NPS, which require subjective grouping of foods and beverages into multiple categories that use differing algorithms and/or scoring thresholds and have trouble scoring mixed meals that contain ingredients across two or more of these categories[15].

## Strengths

This investigation examined nationally representative data on nearly 50,000 U.S. adults, increasing generalizability and statistical power. Dietary data were calculated using up to two 24 hr recalls per individual, allowing precise scoring of individual food and beverage products. While some prior NPS validation studies utilized detailed product information from 24 hr recalls[39–44], others utilized food frequency questionaries[45–48] that only allow scoring of major food categories (e.g., fruits, vegetables, red meats, poultry, eggs, salty snacks, breakfast cereals), which does not truly validate the ability of the NPS to discriminate healthfulness of the diverse food and beverage products within each of these categories. Both the FCS and i.FCS were weighted by energy (kcal), rather than weight or portion size, which prevents bias or the need for differing scoring cut-points for different food and beverage categories due to differences in water weight, fiber, or fat content.

Our endpoints were clinically relevant and well measured, including dietary quality assessed using a validated diet pattern score; major clinical risk factors and prevalent diseases, using standardized physical examination and laboratory measures; and all-cause mortality, prospectively using the NDI. Qualitatively, we evaluated and demonstrated favorable associations with a greater number of cardiometabolic health outcomes than any prior NPS validation study to date. All analyses adjusted for a range of well measured demographic, socioeconomic, and lifestyle factors, reducing confounding. Relationships with mortality were prospectively assessed, establishing temporality. Findings were similar in several sensitivity analyses, increasing confidence in robustness of the findings.

## Limitations

Cross-sectional analyses preclude assessment of temporality; however, findings were consistent in prospective analyses of all-cause mortality. Use of energy-weighting to calculate i.FCS captures overall dietary composition but provides lower weighting to certain foods with lower calories per servings, such as fruits and vegetables. However, portion- or gram-based weighting would create bias by failing to account for water weight, as described above. For practical goals of creating a single NPS score, some simplifications were required, such as using the RDA for 19–50 year-old men as target thresholds for several nutrients. In the future, a more "personalized" FCS could be crafted based on specific characteristics of the individual, such as age, sex, disease status, and more. Some nutrients such as Vitamin D, choline, and flavonoids were only available in certain NHANES cycles, requiring imputation in other cycles. Because errors in nutrient imputation would unlikely be systematic with respect to disease outcomes or total mortality, this imputation may have attenuated findings toward the null. While using two 24 hr recalls provides a reasonable measure of an individual's diet, within-person dietary variation reduces the ability to detect associations with clinical risk factors and health outcomes[49]. Misreporting and omission of food items by dietary recall participants was also possible. If such error were random with respect to the outcomes, this would attenuate findings toward the null. If systematic with respect to the outcomes, for instance if individuals at higher health risk were more likely to underreport unhealthy foods and beverages, this would artificially inflate these individuals' i.FCS – a bias which would weaken the reported findings and make the i.FCS appear less protective than it actually is.

Certain fasting biomarker levels (e.g., triglycerides, fasting plasma glucose) were missing for a significant proportion of the NHANES sample, which may have weakened the accuracy of the estimates for these health indicators. However, assessment of complementary biomarkers with less missingness for long-term glucose homeostasis (HbA1c) and blood lipid profiles (HDL-C, total cholesterol: HDL ratio) yielded generally consistent findings. The associations of i.FCS with diet quality, clinical risk factors, and mortality are observational, and residual confounding cannot be excluded. Yet, associations remained significant after adjustment for a range of well-measured covariates, and analyses of individual domains are consistent with known biologic effects of several of the attributes within these domains. Finally, we did not compare the findings to other NPS, which can be done in future work.

The Food Compass, extended based on scores of specific food and beverage products to a person's dietary intake, associates with a healthy dietary pattern, major clinical risk factors and prevalent conditions, and longitudinal risk of all-cause mortality. These findings support the validity of Food Compass as a potential tool to guide public and private strategies to identify and encourage healthier foods and beverages.

## Methods

### Study design and participants

We utilized nationally representative data among 47,999 adults aged 20–85 years enrolled in the 10 most recent cycles of NHANES, 1999–2000 through 2017–2018 (Fig. S1). NHANES uses trained personnel and standardized methods including in-person physical examination to collect information on sociodemographic factors, clinical risk factors, health conditions, and dietary and other lifestyle habits from recurrent cross-sectional nationally representative samples. In addition, through the most current available linkage to the National Death Index (NDI), we prospectively evaluated mortality outcomes among 47,918 NHANES adult participants from the 1999–2000 to 2017–18 cycles (excluding 81 participants who did not have linked NDI information). All analyses accounted for the complex survey design and sampling weights to be representative of the non-

institutionalized U.S. population. NHANES was approved by the Institutional Review Board at the National Center for Health Statistics (NCHS), and all participants provided informed written consent, and were provided compensation and a report of medical findings.

### Dietary assessment

As part of the What We Eat in America (WWEIA) component of NHANES, dietary intake was assessed using standardized 24 hr dietary recalls, administered by trained NHANES interviewers, using United States Department of Agriculture's (USDA) Automated Multiple Pass Method[50]. A single 24 hr recall was completed in 1999–2000 and 2001–2002, and a second 24 hr recall was added for most respondents (87%) from 2003–2004 onwards. We used all complete and reliable dietary records available[51] (Supplementary Note 1), as determined by the National Center for Health Statistics staff, averaging the two recalls for each individual when present to reduce the influence of day-to-day dietary variation.

Product-level nutritional information was extracted from the Food and Nutrient Database for Dietary Studies (FNDDS) from 2001–2018, food ingredients data from the Food Pattern Equivalents Database (FPED), and flavonoid data from the 2010 USDA Flavonoid database. We excluded infant formula, baby foods, alcohol, specialized medical foods and dietary supplements, and items providing < 5 kcal per 100 g (e.g., unsweetened tea or coffee, diet soda). Any missing attributes (i.e., nutrient quantities) required for nutrient profiling were estimated by first propagating attribute values for an item with an identical food code available in the most recent survey cycle; and then for the remaining, by predictive mean matching (PMM) imputation using FNNDS and FPED attribute values and WWEIA indicator category (152 categories) as predictors for the food item (Supplementary Note 2, Table S11). PMM imputation fills in a value randomly from among the observed donor values from an observation whose regression-predicted values are closest to the regression-predicted value for the missing value from a simulated model to ensure the imputed values are plausible, particularly when the normality assumption is violated[52,53].

### Food Compass scoring

Briefly, Food Compass incorporates 54 attributes across 9 domains, namely: nutrient ratios, vitamins, minerals, food ingredients, additives, processing, specific lipids, fiber and protein, and phytochemicals (Tables S1–S2)[15]. Each attribute is scored from 0 to 10 for beneficial attributes, −10 to 0 for harmful attributes and −10 to 10 for ratios, with domain scores calculated as an average of its component attributes (or for the food ingredients domain, as the sum). Domain scores are summed, with the latter three domains (e.g., specific lipids, fiber and protein, and phytochemicals) each receiving a half weight; and the summary FCS scaled to range from 1 (least healthful) to 100 (most healthful). The algorithm included several minor modifications, identified since publication of our original report, such as a more graded approach to scoring processed meats and fermented foods (Supplementary Note 3, Table S12). In addition, due to data unavailability in FNDDS, seven attributes were excluded from Food Compass scoring (Table S1). The same attribute scoring and algorithm are used across all food and beverage products. Items with a score of ≥ 70 were generally identified as foods to be encouraged; with a score of 31–69, foods to be eaten in moderation; and with a score of ≤30, to be minimized[15].

To extend FCS for specific products to an individual's overall diet, the FCS score for each item reported in a person's diet was weighted by its percent contribution to that person's total energy intake, and then summed, to calculate an individual's Food Compass Score (i.FCS). Because alcohol is not scored by FCS, energy from alcohol intake was excluded from the i.FCS calculation and included as a covariate in all models. The i.FCS for each individual could theoretically range from 1 to 100.

To understand how different combinations of items could contribute to a similar i.FCS, we assessed the distribution (number and contribution to total energy intake) of consumed food and beverage products, based on previously defined product healthfulness thresholds of FCS ≤ 30, 31–69, and FCS ≥ 70, among subgroups of the US populations having various i.FCS.

To explore the contribution of each separate Food Compass domain to dietary quality and health associations, we also calculated an energy-weighted individual dietary score for each of the 9 FCS domains, using similar methods, and also each ranging from 1 to 100. The i.Domain scores were not used to derive the i.FCS, but separately calculated from the product-level domain scores to explore their associations with the endpoints of this investigation.

### Sociodemographic factors and covariates

In NHANES, sociodemographic factors including age, sex, race/ethnicity, household income, and education levels were assessed by trained personnel. Race/ethnicity was self-identified by each participant based on standardized response categories to questions pertaining to race and Hispanic origin: non-Hispanic White, non-Hispanic Black, Mexican-American, other Hispanic, and Asian or other race, including multiracial. The ratio of family income to poverty level was calculated by dividing the participant's self-reported family income by the poverty guidelines specific to family size, appropriate year, and state. Additional lifestyle covariates included in the models were: smoking status, total physical activity, average total energy intake, and alcohol consumption. Smoking status was categorized as never, former, or current smoker based on a range of questions regarding the respondent's self-reported smoking habits. Total physical activity was assessed continuously as self-reported Metabolic Equivalent of Task (MET)-hours per week. Methodological differences in how physical activity was assessed in NHANES from 1999 to 2006 and from 2007 to 2018 preclude the use of a consistent variable for total physical activity across survey cycles. As such, we used a total physical activity variable (MET-hours/week) evaluated from 2007 to 2018, a leisure time physical activity variable (MET-hours/week) for all survey cycles, and all other demographic and health data to impute total physical activity (MET-hours/week) for 1999–2006 survey cycles using PMM (Supplementary Note 4, Table S13–15). The relative energy contribution of each food item was accounted for in the dietary i.FCS calculation. Total energy intake was not included as a component of either the FCS (food level) or the i.FCS (dietary level). Thus, we adjusted for total energy intake derived from the 24 hr recall data available and adjusted for it continuously as a potential confounder in the models. Alcohol consumption was also modeled continuously as the percentage of average total energy intake from alcohol.

### Outcomes

i.FCS was validated against three different outcomes: (a) the Healthy Eating Index (HEI) 2015, a validated measure of a healthy dietary pattern; (b) prevalence of major clinical risk factors and health conditions; and (c) prospective risk of all-cause and cause-specific mortality.

**Healthy dietary pattern.** We calculated HEI 2015 for each individual, a measure based on adherence to the 2015–2020 Dietary Guidelines for Americans (DGAs). HEI 2015 is calculated based on key food groups and nutrients emphasized in the DGAs, resulting in a score ranging from 0 to 100. HEI 2015 has been widely validated against health outcomes, including all-cause and cardiometabolic mortality, cardiometabolic health outcomes, and cancer[16–19].

**Clinical risk factors and prevalent health conditions.** We assessed major clinical risk factors in NHANES including body mass index (BMI) (weight in kilograms divided by height in meters squared); systolic and diastolic blood pressure (mm Hg); HDL-cholesterol, LDL-cholesterol,

triglycerides (mg/dL), and the total cholesterol: HDL ratio; and Hba1c (%) and fasting plasma glucose (mg/dL). For respondents with missing biomarker values (i.e., triglycerides, LDL cholesterol, and fasting plasma glucose were only collected on a fasting sub-sample of each NHANES cohort), PMM imputation was used to resolve missing values using all available health and demographic variables as predictors (Supplementary Note 5, Table S16). We also assessed health conditions, including the presence of metabolic syndrome, diabetes, cardiovascular disease (CVD), cancer, and lung disease; as well as optimal cardiometabolic health, characterized by optimal levels of adiposity, blood glucose, blood lipids, blood pressure, and absence of prior clinical CVD (Table S3). Missing values for self-reported health conditions and medications were assumed to be normal (i.e., no prevalent condition or medication use).

**All-cause and cause-specific mortality.** NHANES survey data were linked to the National Death Index (NDI) mortality data from the Public-Use Linked Mortality Files[54], allowing prospective assessment of mortality through Jan 1, 2019. The NDI captures ~97% of deaths when social security numbers are available, which is the case for all eligible WWEIA respondents linked to the NDI[55,56]. NCHS staff screen respondents from NHANES for identifying information such as social security number, name, date of birth, and state of residence, and use probabilistic matching methods to link these respondents to records in the NDI. Deaths were adjudicated by the NCHS using standardized procedures. Our primary endpoint was mortality from all causes with secondary outcomes of deaths from cardiometabolic disease (sum of deaths from coronary heart disease, stroke, and diabetes) and from all cancer.

### Statistical analysis

The relationship of i.FCS with HEI 2015 was assessed using Spearman correlation. To assess relationships with clinical risk factors, prevalent health conditions, and mortality outcomes, i.FCS was evaluated per each standard deviation (SD) unit. We used survey-weighted linear regressions with robust standard errors to estimate the association of i.FCS with continuous risk factors; and survey-weighted logistic regression to estimate the odds ratio (OR) and 95% confidence interval (CI) for the association with prevalent health conditions. Each FCS domain was similarly evaluated. Multivariable models were used to minimize potential confounding, with covariates included based on clinical and biological relevance to both exposures and outcomes, including age and age-squared (continuous), sex, race/ethnicity (non-Hispanic White, non-Hispanic Black, Mexican-American, Asian/other), education (< high school graduate, high school graduate, some college education, college graduate or more), income (based on family poverty: income ratio, continuous), smoking status (non-smoker, former, current), alcohol use (percentage energy contribution from alcohol, continuous), total physical activity (MET-hours per week, continuous), and self-reported prevalent diabetes, which was identified as being significantly cross-sectionally associated with dietary habits to reduce added sugar. Further adjustment for self-reported CVD and cancer had no appreciable effect on results. In sensitivity analyses, we further adjusted for total energy intake, which could be a confounder or mediator of the relationship between i.FCS and these outcomes, and for NHANES survey cycle as an indicator variable.

Survival analyses used survey-weighted Cox proportional hazard models to estimate the hazard ratio (HR) and 95% CI between i.FCS and risk of all-cause and cause-specific (cardiometabolic and cancer) mortality. Study time was used as the time variable. Participants contributed person-time from the time of their enrollment until death or end-of-follow-up on January 1, 2019, whichever came first. We censored (e.g., removed as events) early deaths within 3 months of baseline, to minimize reverse causation. The proportional hazards assumption was assessed with visual Kaplan-Meier plots and statistical testing of

Schoenfeld residuals (and not rejected if $p > 0.05$), adjusted for age and age-squared (continuous). Multivariable models were adjusted as described for the cross-sectional analyses. In addition, a "mediator" model was assessed in which each of the following could be either confounders or mediators (i.e., in the downstream causal pathway) of the relationship between i.FCS and mortality: major risk factors (BMI, SBP, DBP, HDL, LDL, TG, TC:HDL ratio, HbA1c, FPG; all continuous), self-reported health conditions (CVD, cancer, lung disease), and medications (for diabetes, hypertension, lipid, and angina). In sensitivity analyses, we additionally censored deaths at older ages (85 + years), and further adjusted for total energy intake and NHANES survey cycle as an indicator category. We also assessed potential nonlinear relationships using semi-parametric restricted cubic splines, excluding the top and bottom 1% of i.FCS values and using the $5^{th}$ percentile of i.FCS as the reference. In exploratory analyses, we investigated the relationship between i.FCS and total mortality in subgroups by age, sex, race/ethnicity, education, income, BMI, and plausibility of energy reporting to assess potential variation (interaction) in the association according to these key sociodemographic factors.

All analyses incorporated NHANES dietary sample weights to account for the complex survey design and response rates, and provide nationally representative estimates for the non-institutionalized U.S. population.

### Reporting summary

Further information on research design is available in the Nature Research Reporting Summary linked to this article.

## Data availability

The attribute and domain scoring algorithm used to generate the Food Compass is available in Supplementary Table S2. All data used in this analysis is publicly available from the following USDA and CDC sources: (1) Nutrient composition data for foods reported in NHANES dietary recalls [USDA Food and Nutrient Database for Dietary Studies 2001-2018, FNDDS: https://www.ars.usda.gov/northeast-area/beltsville-md-bhnrc/beltsville-human-nutrition-research-center/food-surveys-research-group/docs/fndds-download-databases/]. (2) Food ingredients data for foods reported in NHANES dietary recalls [USDA Food Pattern Equivalents Database, 2001-2018: https://www.ars.usda.gov/northeast-area/beltsville-md-bhnrc/beltsville-human-nutrition-research-center/food-surveys-research-group/docs/fped-overview/]. (3) Flavonoid data for select foods reported in NHANES dietary recalls [USDA Flavonoid Database, 2007-2010: https://data.nal.usda.gov/dataset/usda-database-flavonoid-content-selected-foods-release-32-november-2015. (4) National dietary recall, sociodemographic, physical activity, smoking, cardiometabolic biomarker, and prevalent condition data for US adults [National Health and Nutrition Examination Survey 1999-2018, NHANES: https://wwwn.cdc.gov/nchs/nhanes/continuousnhanes/default.aspx. (5) All-cause and cause-specific mortality data for US adults [National Death Index 1999–2018, NDI: https://www.cdc.gov/nchs/data-linkage/mortality-public.htm.

## Code availability

Custom code was developed using R (Version 4.0.3) and Stata SE (Version 15.1) for data cleaning, adjustments, and merging, algorithm scoring of FCS for individual food and beverage items, i.FCS scoring of individuals, cross-sectional and survival analyses of i.FCS with health outcomes, and summary statistics, tables, and figures. Statistical significance was defined as two-tailed alpha=0.05. Tufts University is considering licensing of expertise, knowledge, and background work for potential commercial viability of Food Compass for the private sector and non-profit applications, and thus the code is not publicly available. There are no intellectual property or patent protections associated with Food Compass. The detailed Food Compass algorithm and it scoring are specified in our earlier Nature Food Publication[15], as

well as in Table S2 of the current supplementary materials, and can be reproduced freely by any individual or organization.

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

## Acknowledgements

The authors are grateful to Lu Wang, PhD (Friedman School of Nutrition Science and Policy, Tufts University) for providing NOVA classification scores for FNDDS products (2001–2018); Mengxi Du, MS RD (Friedman School of Nutrition Science and Policy, Tufts University) for assistance with the smoking variable in NHANES 1999–2018; and Mengyuan Ruan, MS (Mass General Bringham) for assistance with the physical activity variables in NHANES 1999–2018. This research was supported by the National Institutes of Health (2R01HL115189) and Vail Innovative Global Research (grant N316001 PR0677), both awarded to D.M. The funders had no role in the design or conduct of the study; collection, management, analysis, or interpretation of the data; the preparation, review, or approval of the manuscript; or the decision to submit the manuscript for publication.

## Author contributions

All authors had full access to all of the data in the study and take responsibility for the integrity of the data and the accuracy of the data analysis. M.O. and D.M. were responsible for the concept and design of the analysis. M.O., J.E.M., B.N.L., S.G., and D.M. acquired and interpreted the data. M.O. conducted data and statistical analyses. M.O. and D.M. drafted the manuscript. J.E.M., S.G., C.E., J.B.W., J.B.B., and D.M. critically reviewed the manuscript for its intellectual content. D.M. obtained funding and supervised the entire body of work.

## Competing interests

Ms. O'Hearn reports research funding from the National Institutes of Health and Vail Innovative Global Research. Dr. Erndt-Marino reports income from Bespoke Analytics, LLC, which received research funding from the Florida Department of Citrus, outside the submitted work. Ms. Gerber reports research funding from the National Institute of Food and Agriculture (USDA) and the National Institutes of Health, and additional income from Inova Medical Systems in the Beatty Liver and Obesity Research Program as well as the USDA-Tufts Human Nutrition Research Center on Aging, all outside the submitted work. Ms. Lauren has received personal fees from Abt Associates and the Centers for Disease Control and Prevention, both outside the submitted work. Dr. Economos reports research funding from the United States Department of Agriculture National Institutes of Health, JPB Foundation, and Newman's Own Foundation. She also reports her position as Vice Chair to National Academies of Science Roundtable on Obesity Solutions (unpaid) and her Advisory Board position at Care/of Scientific. None of the above relate to the manuscript. Dr. Wong reports research funding from the National Institutes of Health and membership in the US Preventive Services Task Force (unpaid) and National Academies of Sciences, Engineering and Medicine Committee on Evaluating the Process to Develop the Dietary Guidelines for Americans, 2020-2025 (unpaid), outside the submitted work. Dr. Blumberg reports being on the scientific advisory board for Bragg Live Food Products, LLC, California Prune Board, California Walnut Commission, Cranberry Institute, Good Pharma, LLC, Guiding Stars Licensing Co., Inside Tracker/Segeterra Inc., and January.ai, also all outside the submitted work. Dr. Mozaffarian reports research funding from the National Institutes of Health, Gates Foundation, Rockefeller Foundation, and Vail Innovative Global Research; and scientific advisory board for Beren Therapeutics, Brightseed, Calibrate, DayTwo (ended 6/2021), Elysium Health, Filtricine, Foodome, HumanCo, January Inc., Perfect Day, Season and Tiny Organics, all outside the submitted work.

## Additional information

**Correspondence and requests** for materials should be addressed to Meghan O'Hearn.

