## [Peer Review File · Nature Communications]

REVIEWER COMMENTS

Reviewer #1 (Remarks to the Author):

Mike Rayner, University of Oxford

General comments

This is an interesting paper describing analyses aimed at validating the Food Compass nutrient profile model against dietary and health outcomes in a sample of US adults and I recommend that it should be published by Nature Food This is primarily because Nature Food have already published a paper describing the development of the Food Compass nutrient profile model (Mozzafarian et al, 2021).

In reviewing that previous paper I said that: 'The authors say [page 21 line 474] 'important next steps include testing and validation against health outcomes' There are now to my knowledge at least four nutrient profile models that have been validated against health outcomes to varying extents (the UK FSA/Ofcom model, Nutri-Score, HSR and ONQI). I would strongly suggest that the Food Compass Score should be validated against health outcomes in comparison with at least one of these other models. This should probably be done before publication of this paper and the results published in parallel with it'.

The present paper by O'Hearn et al does present the results of testing the Food Compass nutrient profile model against health outcomes. This is significant progress. But the authors have yet to do this in comparison with any other nutrient profile model that has been validated against health outcomes and has been shown to be valid (in particular the Nutri-Score algorithm). I recommend that the authors state this clearly in this paper.

Until then I do not think the authors are justified in claiming superiority for their model over other nutrient profile models and indeed other food classification systems. Their claim that 'the Food Compass could be a possible unified standard NPS' [Line 396] for efforts to develop unified front-of-pack (FOP) label in the EU and elsewhere is both unjustified and unrealistic and should be qualified.

Besides demonstrating predictive validity of the nutrient profile model in relation to healthy dietary patterns and improved health outcomes. This paper also explored how each of the 9 scoring domains of the Food Compass Score separately related to dietary and health outcomes [Line 86] but the authors barely explore the implications of their findings here. This paper would be a much more interesting paper if it did explore some, at least, of those implications. Some commentary here (if not further analyses) in relation to the contribution of domain scores to final Food Compass cores is recommended.

The authors show, for example, that ‘the i.FCS for individuals was highly correlated with HEI-2015 (R=0.82) (Figure 2)’ but that: ‘Individually, the 9 domains of the FCS were not as highly correlated, ranging from 0.16 for i.Specific Lipids to 0.71 for i.Nutrient Ratios (Figure S2)’ [Line 268]. So doesn’t this mean that using the i.Nutrient Ratios domain for the i.FCS gives you almost all you need to give you a ‘good enough’ ranking of diets at least compared with the HEI 2015? What then is the added benefit of including, for example, the i.Specific Lipids domain in the total i.FCS and by extrapolation the Food Compass Score for foods and can this be quantified?

This matters because the authors continue, in my view, to fail to justify the inclusion of 54 components in the FCS when most other nutrient profile models manage with much fewer and also because the information needed to score many of the components of the Food Compass nutrient profile model is unavailable in many food composition databases other than that used by the authors in this paper. E.g. trans fatty acid information is commonly unavailable in many food composition databases and so too is the information necessary to score foods for degree of processing (here according to the NOVA classification system).

Similarly the authors show significant inverse associations with all-cause mortality for seven of the nine domains of the i. Food Compass Score [Line 329], i.e. for Nutrient Ratios, Vitamins , Additives Processing, .Fiber and Protein, Specific Lipids and Phytochemicals and not for Minerals and Food Ingredients [Table S8]. Again does this not suggest that the Mineral and Food Ingredients domains could usefully be omitted from the final score?

Specific comments

Line 74. ‘The Food Compass is a novel NPS that incorporates a range of 54 protective and risk factor nutrients, ingredients, bioactives, additives and processing attributes, grouped across 9 domains, and selected and weighted based on the latest evidence about their relative healthfulness. The Food Compass Score FCS), ranging from 1 (least healthful) to 100 (most healthful), enables a more consistent and universal algorithm across all food and beverage categories; permits similar scoring for mixed dishes and meals; and improves convergent and discriminatory validity of product scoring compared to other major NPS.

The comparative novelty of the Food Compass system does indeed mainly lie in the number and type of food attributes that are encompassed but in my view provides no more consistent scoring or convergent validity than other nutrient profile models. Furthermore I continue to fail to see why having more components than other nutrient profile models ‘enables a more consistent and universal algorithm

across all food and beverage categories' as the authors claim. But I have said this before in my review of the previous paper by Mozzafarian et al and the criticism seems to have gone unheeded.

Line 162 'To extend FCS for specific products to an individual's overall diet, the scores for each item reported in a person's diet were summed, weighted by its percent contribution to that person's total energy intake and then used to calculate an individual's Food Compass Score (i.FCS).' There are many different ways of converting nutrient profile scores for foods into diet quality scores for individuals [e.g. see <https://discovery.ucl.ac.uk/id/eprint/1369569/1/Thesis-GMASSET-UCL-2012-FINAL.pdf>] It would be useful if the authors could explain their choice at greater length (e.g. as they say [Line 430] 'Use of energy-weighting to calculate i.FCS provides lower weighting to certain foods with lower calories per servings such as fruits and vegetables. It seems to me to be likely that the particular way nutrient profile model scores should be converted to diet quality scores depends on the use to which the nutrient profile model is put (e.g. it will be different, say, if the model is for front-of-pack labelling purposes as opposed to restrictions on the marketing of foods to children)

Line 416 'in contrast, several validation studies of other NPS (40-43) utilized food frequency questionnaires that only allow scoring of major food categories.' This is a selective comparison. There are many other validation studies of nutrient profile models that have used dietary data from multiple 24-hour recalls (e.g. validation studies of the NutriScore algorithm involving the NutriNet Sante cohort and the SUVIMAX study. (e.g https://academic.oup.com/eurpub/article/30/Supplement_5/ckaa166.1285/5914738?login=true

Reviewer #2 (Remarks to the Author):

This paper follows on from a publication by the same group in Nature Foods in 2021 which described a new nutrient profiling score (Food Compass). This tool (like other NPSs) is aimed at assessing the healthfulness of individual foods. This perspective of individual foods links this NPS (and others) to the issue of front of pack labelling.

The purpose of this new paper is different as it aims to take the Food Compass score for each food eaten by an individual and then to summate it to provide a summary overall score. This is intended to be an

indication of the healthfulness of the diet than individual foods. What the authors show is in an analysis of NHANES that this score relates to various risk factors and prevalent disease in a way that one might expect and that it is predictive of overall mortality in a follow up analysis over nearly 17 years in which there were nearly 5000 deaths in the cohort of 38K individuals included in the prospective analysis.

This analysis does suggest that the individual Food Compass Score does have predictive validity for total mortality in a US population.

There are some minor points as below about the analysis but the major question is not about whether the iFCS has predictive validity or not but rather whether that has any utility either for public health, for clinical practice or for the people themselves.

The authors state that "NPS like Food Compass can be one important tool in such interventions"... (to improve diet quality)... "allowing consumers, industry including (food manufacturers, retailers, and restaurants,), investors, schools, hospitals, and worksites, and policy makers to identify and shift toward healthier food and beverage options". Whilst one can understand how front of pack labelling using some form of NPS might influence individual food choices, it is rather difficult to see how the computation of an overall dietary score using the iFCS would have an impact. The argument for how computation of the iFCS might actually be used, by whom and for what purpose, needs to be much clearer. It is not reasonable to merely state the argument for the individual food NPS because that is not what this paper is about. A clearer explanation of the usefulness of an overall diet NPS is required.

The basis for the calculation of the iFCS is both somewhat arbitrary and very complicated. Why the different domains are equally weighted with the exception of three which receive 0. weights is not explained. Why only the top 5 vitamins and minerals and the top 3 specific lipids are included is not self-evident.

The correlation with the Healthy Eating Index is 0.82. This does beg the question about whether the iFCS has any advantage over the HEI. It is certainly not easier to compute. Some discussion of their relative merits is warranted.

The iFCS quintiles appear to be strongly inversely related to total energy intake and there is a weighting for each food stuff on the basis of their contribution to total energy and it appears at least in some of the analyses that TEI was also included as a covariate. A clear statement about what role TEI plays in this score would be helpful.

There is a lot of missing data in this study. Some variables e.g fasting glucose, LDL-cholesterol and triglycerides are missing in more than 50% of people. In such a situation of extreme missingness, imputation with whatever method is problematic. There are also some challenges with physical activity which was assessed differently in different time periods. In this case one can understand how the use of the different measures of PA can be used to impute missing values. It is less clear how nutrients that are not assessed at all (as far as I understand it) e.g total flavenoids, vitamin D and choline can be estimated by imputation.

There is an important typo in the description of the means which states that the mean HbA1c in the population is 6%. I think this should read 5.6%.

Reviewer #3 (Remarks to the Author):

This manuscript describes an elaborate food scoring system, which has been extended to assessment of an individual's dietary intake. Such scoring ensures like-with-like comparisons and may therefore be used to estimate food quality at point of purchasing as well as diet as a whole. Here, the authors validate the score by cross-sectional as well as prospective indicators of (ill) health. These associations were observed to be largely significant though relatively small in effect size and have plausible biological mechanisms with the domains of the food compass score.

The algorithm behind the food compass is extensive and "not for the faint-hearted". It captures a large range of dietary exposures which otherwise are covered in separate scores. An extensive and elaborate work which emphasizes the complexity and inter-relationships of foods which make up our diet.

My comments are divided in two parts. The first part relates to methodology/interpretation, the second part are just suggestions for the text.

Introduction

Line 51: Please include references after the first sentence.

Line 55: "industry reformulation" or is meant "food reformulation by the industry"?

Methods

Line 93: Please include the upper age range.

Line 109: Which criteria were applied to assess completeness and validity of the diet record data?

Line 116: Why are alcoholic beverages excluded from the score?

Line 149: The manuscript mentions 54 attributes, but Table S1 indicates that certain attributes are not used in the score due to unavailability of data in the various composition tables. Trans fats are marked as not included, but appear in the various tables. Text S1 may also need checking for similar reasons.

Line 158: It would be good to know more about the motivation for these cut-off values (<30, 30-70, >70), what the reasoning behind these has been. Also, when taking this into a public health setting, it would be good to see how the frequency distribution of these three categories is in relation to the i.FCS consumed. In other words, is it about consuming more >70 foods, less <30 foods, or the relative proportion between them?

Line 166: Please clarify that the individual domain scores also range from 0-100.

Line 203: Considering the HEI and the FCS have the same range, apart from showing the scatterplot and correlation between the two scores (figure 2), it would be nice to see the agreement in the form of a Bland-Altman plot.

Results

Line 285: It would be good to have Table S7 in the main text, since it shows the associations for the separate domains and emphasizes their relative importance for the presented risk factors/conditions. Did the results in S7 change much after adjustment for daily energy intake? Considering the domain “additives”, are the observed associations mainly driven by the attribute “added sugar”? Also, protein and fiber are positively associated with prevalent diabetes, may this be due to reverse causality?

Line 274/294/313: Considering the wide age range (20-85+) in the sample, do the authors consider the associations between the i.FCS and various outcomes to be the same across the whole sample?

Figures

Figure 1: What is the reason for the ‘unevenness’ in the distribution for additives?

Figure 2: The Y-axis ends on 00, but this is presumably 100.

Figure S3: This figure is not referred to in the text. It models the domains individually in their association to all-cause mortality (mutually adjusted for). How are these positive and negative associations interpreted/explained?

Tables

On various occasions in the S-tables, the zero (?) behind the decimal point is missing.

Table 2: Could total cholesterol be added?

Table S1: Could an additional column (as first column) be added which indicates 'included' or 'excluded' from the FCS, thereby shifting any attributes listed but not included in the i.FCS into the row for 'excluded'.

Table S2: What is meant by 'total' behind some of the nutrients? Is this food and supplement sources combined? Or might this be e.g. a-TE instead of mg tocopherol. The footnote describes 'original score', what is meant by this?

Table S6: adjustment for daily energy intake and survey cycle minimally changed the associations observed, apart from triglycerides and LDL-C. Just curious whether there is an explanation.

Discussion

Line 406: I agree this is a big advantage. However, the ability of the FCS to capture the various settings in which people eat (home, take-away, restaurant etc) requires a dietary assessment method (and extensive food list) which equally needs to be able to capture and differentiate on these aspects.

Line 417: Even though the foods in the food composition table can be precisely categorized, the i.FCS may still suffer from systematic error due to misreporting and omission of food items by participants. Could the authors refer to the implications for this in the section on limitations?

Line 420: A very good point to make!

*Food composition tables may not contain all the details required to calculate the FCS/i.FCS. How do the authors see the extension of their work to other countries/settings? What would be their advice if certain attributes are missing?

*For the nutrient scores, 25% of the RDAs for men 19-50 years were used, overestimating the domains for certain age-sex groups (e.g. iron, calcium). For intention of use as FoP labelling, this systematic error seems preferable to detail and clutter, but as i.FCS, would a more 'individualized' score taking age and sex into account when it comes to RDA be preferred perhaps? Have alternative algorithms been tried?

*The impact of food production and consumption on the environment is becoming increasingly important. A process has been started for food guidelines to include aspects of sustainability. Could this be the 10th domain? Or would sustainability (like alcohol) be best modelled 'outside' the FCS?

References

Please include an 'accessed date' for the web-based sources.

Ref 44: Could the authors include the chapter or possibly even refer to the original publication(s)?

Textual

Line 25: Suggest removing “consistent”, since this is implied by “algorithm”.

Line 34: Suggest adding “In cross-sectional analysis, [after multivariable ...]”

Line 35: Suggest to remove “levels of” in front of BMI.

Line 38: For better readability add “as well as, [lower prevalence]”

Line 73: Suggest rephrasing to “health risk related nutrients, ...”.

Line 80: Suggest rephrasing to “...and the corresponding validity for the association between a person’s FCS and health outcomes, ...”.

Line 98: Suggest to include “...(NDI) up to 2015 we ...”.

Line 117: Suggest including “Any missing attributes (i.e. nutrient quantities) required ...”

Consider to move the FCS paragraph after the dietary assessment, followed by the paragraph on socio-demographics.

Line 162: Suggest to write the sentence in the order of actions “..., the FCS score for each food item reported in a person’s diet was weighted by its percent ...intake and then summed.”

Line 164: Consider to remove duplicated sentence on alcohol.

Line 210: Suggest to include: “In cross-sectional analysis, all models were adjusted ...”

Line 232: typo casual.

Line 260: Suggest “With increasing i.FCS, the percentage of participants with Asian or other ethnicity increased; whereas the reverse was observed for Non-Hispanic Black participants.”

Line 270: Suggest to add “...highly correlated with the HEI-2015, ranging from ...”

Line 274: Suggest to change to: “In multi ..., higher i.FCS were significantly associated with more favorable risk factors (Table 2).”

Line 297: To make the difference with the survival analysis clearer, suggest to change to: “...was associated with 15% lower odds of having the metabolic syndrome, ...etc”

Line 343: ‘this NPS’ refers to the FCS.

Line 387: Check location of brackets.

Line 401: Would be good to explain more, to avoid misinterpretation of black box.

Line 416: Suggest to add “using 1-2 24-hour diet recalls per individual ...”.

Table S4: The numbers after the decimal for kcal could be removed.

Tufts University Friedman School of Nutrition Science and Policy
150 Harrison Avenue
Boston, MA 02111

June 10, 2022

Response to Reviewer Comments

Thank you to the reviewers for their thoughtful feedback on our original submission, “**Validation of the Food Compass Score with a healthy diet, cardiometabolic health, and mortality among U.S. adults, 1999-2018.**” We are pleased to submit our revised manuscript, with careful changes in response to these comments. Our detailed responses to each comment are summarized below.

In addition, we have made the following analytical updates:

(1) inclusion of National Death Index mortality data through January 1, 2019 in the prospective survival analyses which increases the number of deaths in the analysis from 4953 to 7481 [data was recently released by NDI]; (2) modification from gram-weighted to energy-weighted scoring of a product’s constituent ingredients for NOVA classification – one of the attributes of the Food Compass algorithm.

We believe the manuscript is strengthened as a result of these suggested revisions and analytical updates, and hope this is now suitable for publication.

Best regards,

Meghan O’Hearn, MS

PhD Candidate, Friedman School of Nutrition Science and Policy, Tufts University

Meghan.o_hearn@tufts.edu

617-417-6411

REVIEWER COMMENTS

Reviewer #1 (Remarks to the Author):

Mike Rayner, University of Oxford

General comments

1. This is an interesting paper describing analyses aimed at validating the Food Compass nutrient profile model against dietary and health outcomes in a sample of US adults and I recommend that it should be published by Nature Food This is primarily because Nature Food have already published a paper describing the development of the Food Compass nutrient profile model (Mozaffarian et al, 2021).

Thank you for these positive comments. We agree this manuscript is well suited for the Nature family of journals, and believe Nature Communications will make an excellent home.

2. In reviewing that previous paper I said that: ‘The authors say [page 21 line 474] ‘important next steps include testing and validation against health outcomes’ There are now to my knowledge at least four nutrient profile models that have been validated against health outcomes to varying extents (the UK FSA/Ofcom model, Nutri-Score, HSR and ONQI). I would strongly suggest that the Food Compass Score should be validated against health outcomes in comparison with at least one of these other models. This should probably be done before publication of this paper and the results published in parallel with it’.

The present paper by O’Hearn et al does present the results of testing the Food Compass nutrient profile model against health outcomes. This is significant progress. But the authors have yet to do this in comparison with any other nutrient profile model that has been validated against health outcomes and has been shown to be valid (in particular the Nutri-Score algorithm). I recommend that the authors state this clearly in this paper.

Until then I do not think the authors are justified in claiming superiority for their model over other nutrient profile models and indeed other food classification systems. Their claim that ‘the Food Compass could be a possible unified standard NPS’ [Line 396] for efforts to develop unified front-of-pack (FOP) label in the EU and elsewhere is both unjustified and unrealistic and should be qualified.

Shown previously superiority for indentifying refined starches, healthy fats – mozaaffarian 2021
“could be a possible”

In the first paper on Food Compass, we assessed three major domains of validity: content validity, by assessing nutrients, food ingredients, and other characteristics of public health concern; face validity, by assessing FCS for 8,032 foods and beverages reported in NHANES/FNDDS 2015-16; and convergent and discriminant validity, from comparisons to NOVA food processing classification, Health Star Rating, and Nutri-Score. Here, we present detailed findings on the fourth domain of validity: construct validity, evaluating Food Compass against population diet quality indices and health outcomes.^{13,14} With these aims, we first applied specific product scores to a person’s diet, deriving an energy weighted individual Food Compass Score (i.FCS). The present investigation then performed several important analyses, including assessment against the Health Eating Index, assessment against health risk factors and prevalent disease conditions, and assessment against risk of total mortality. We also performed analyses of these complex health endpoints for each of the nine domains of the Food Compass separately, providing novel information on how the different domains relate to disease risk.

We agree that separate, future analyses could compare different nutrient profiling systems in relation to health risk. These are not simple analyses, for example requiring formal consideration of potential differences in discrimination (e.g., area under the ROC) and calibration (e.g., risk reclassification), and methods for testing statistical significance of these differences. Such analyses are beyond the scope of the present manuscript, which aims to validate the Food Compass construct validity, against a healthy eating pattern, against health risk factors and prevalent disease conditions, and against risk of total mortality.

We also note that requiring formal comparisons against other NPS, a complex separate undertaking, in an initial validation paper is inconsistent with any prior published validation paper on NPS, including several on which this Reviewer is a co-author. In the Table below, we present the 15 prior published validation studies of NPS against health outcomes. None of these compared the association of multiple NPS against health outcomes, but focused on careful validation of one NPS of interest against health outcomes.

As recommended by the Reviewer, we now state in the Discussion that neither we nor others have compared the validity of different NPS against health outcomes, and that this an important area for future work, including assessing potential comparative differences of different NPS in different nations and subpopulations.

We do not state in this paper that the FCS is “superior” to other systems. We do believe this NPS, with its characterization of refined starch, healthy fats, processing, phenolics, additives, nutrient ratios, and multiple food ingredients, “could be” a possible unified standard NPS. It has many strengths, and this should be considered for this purpose.

Table. Prior published investigations assessing the association between individual dietary scores derived from a nutrient profiling system and health outcomes. None of these compared the NPS of interest against another NPS. That is an area for potential future work.

Publication	subjects/dataset	NPS assessed	Association assessed
Streppel et al. 2012 ¹	Rotterdam Study (n=4969)	Nutrient Rich Food Index (NRF9.3)	Association between NRF9.3 dietary index score and anthropometric measures at baseline
Streppel et al. 2014 ²	Rotterdam Study (n=4969)	NRF9.3	Association between NRF9.3 dietary index score and risk of all cause mortality and major CVD incidence.
Chiuvè et al. 2011 ³	Nurses' Health Study [NHS] (62,284 women) and Health Professionals Follow-up Study [HPFS] (n=42,382 men)	Overall Nutritional Quality Index (ONQI)	Association between ONQI dietary index score and total chronic disease (CVD, cancer, diabetes) incidence and all-cause mortality
Deschasaux et al. 2018 ⁴	European Prospective Investigation into Cancer and Nutrition [EPIC] study (n=471,495)	British Food Standards Agency Nutrient Profiling System (FSAm-NPS) *	Associations between the FSAm-NPS dietary index and overall- and location-specific cancer incidence
Deschasaux et al. 2020 ⁵	European Prospective Investigation into Cancer and Nutrition [EPIC] study (n=501,594)	FSAm-NPS*	Associations between the FSAm-NPS dietary index and all-cause and cause-specific mortality
Gomez-Donoso et al. 2021 ⁶	SUN cohort (n=20,503)	FSAm-NPS*	Association between FSAm-NPS dietary index and all-cause, cardiovascular and cancer mortality. Also assessed consistency with dietary recommendations (i.e. Mediterranean Diet Score and Spanish Food pyramid)
Adriouch et al. 2020 ⁷	SU.VI.MAX cohort (n=6515)	FSAm-NPS*	Association between FSAm-NPS dietary index and CVD incidence
Adriouch et al. 2017 ⁸	NutriNet Sante cohort (n=75,801)	FSAm-NPS*	Association between FSAm-NPS dietary index and CVD incidence
Julia et al. 2015 ⁹	SU.VI.MAX cohort (n=3741)	FSAm-NPS*	Association between FSAm-NPS dietary index and Metabolic Syndrome incidence
Julia et al. 2015 ¹⁰	SU.VI.MAX cohort (n=4344)	FSAm-NPS*	Association between FSAm-NPS dietary index and mean BMI change
Lichtenstein et al. 2013 ¹¹	National Health and Nutrition Examination Survey [NHANES] (n=11,296)	AHA Heart-Check Food Certification Program (AHA HCP)	Association between AHA HCP quartile of intake (% E) and cardiometabolic risk factors and conditions
Erkisen et al. 2018 ¹²	Airwave Health Monitoring study (n=5848)	UK Nutrient Profile (NP) Model	Association between NP model and metabolic risk factors for CVD and diabetes
Pan et al. 2020 ¹³	Australian Diabetes, Obesity and Lifestyle Study [AusDiab] (n = 10025)	Health Star Rating (HSR)	Association between HSR and all-cause and CVD mortality
Donat-Vargas et al. 2021 ¹⁴	Study on Nutrition and Cardiovascular Risk in Spain [ENRICA] (n=12,054)	5 color Nutri-Score and continuous FSAm-NPS*, based on g/d/kg and based on percent energy (4 versions total)	association between Nutri-Score variants (5-CNS and FSAm-NPS) and all-cause and cause specific mortality
Mytton et al. 2018 ¹⁵	EPIC-Norfolk study (n=25,639)	FSA-Ofcom model	Association between FSA-Ofcom model and CVD incidence

*The British Food Standards Agency Nutrient Profiling System (FSAm-NPS) is the continuous score underlying Nutri-Score

1. Streppel MT, de Groot LC, Feskens EJ. Nutrient-rich foods in relation to various measures of anthropometry. *Fam Pract.* 2012;29 Suppl 1:i36-43.
2. Streppel MT, Sluik D, van Yperen JF, et al. Nutrient-rich foods, cardiovascular diseases and all-cause mortality: the Rotterdam study. *European journal of clinical nutrition.* 2014;68(6):741-747.
3. Chiuve SE, Sampson L, Willett WC. The association between a nutritional quality index and risk of chronic disease. *American journal of preventive medicine.* 2011;40(5):505-513.
4. Deschasaux M, Huybrechts I, Murphy N, et al. Nutritional quality of food as represented by the FSAm-NPS nutrient profiling system underlying the Nutri-Score label and cancer risk in Europe: Results from the EPIC prospective cohort study. *PLoS Med.* 2018;15(9):e1002651.
5. Deschasaux M, Huybrechts I, Julia C, et al. Association between nutritional profiles of foods underlying Nutri-Score front-of-pack labels and mortality: EPIC cohort study in 10 European countries. *BMJ (Clinical research ed).* 2020;370:m3173.
6. Gómez-Donoso C, Martínez-González M, Perez-Cornago A, Sayón-Orea C, Martínez JA, Bes-Rastrollo M. Association between the nutrient profile system underpinning the Nutri-Score front-of-pack nutrition label and mortality in the SUN project: A prospective cohort study. *Clin Nutr.* 2021;40(3):1085-1094.
7. Adriouch S, Julia C, Kesse-Guyot E, et al. Prospective association between a dietary quality index based on a nutrient profiling system and cardiovascular disease risk. *European Journal of Preventive Cardiology.* 2020;23(15):1669-1676.
8. Adriouch S, Julia C, Kesse-Guyot E, et al. Association between a dietary quality index based on the food standard agency nutrient profiling system and cardiovascular disease risk among French adults. *Int J Cardiol.* 2017;234:22-27.
9. Julia C, Fézeu LK, Ducrot P, et al. The Nutrient Profile of Foods Consumed Using the British Food Standards Agency Nutrient Profiling System Is Associated with Metabolic Syndrome in the SU.VI.MAX Cohort. *J Nutr.* 2015;145(10):2355-2361.
10. Julia C, Ducrot P, Lassale C, et al. Prospective associations between a dietary index based on the British Food Standard Agency nutrient profiling system and 13-year weight gain in the SU.VI.MAX cohort. *Preventive medicine.* 2015;81:189-194.
11. Lichtenstein AH, Carson JS, Johnson RK, et al. Food-intake patterns assessed by using front-of-pack labeling program criteria associated with better diet quality and lower cardiometabolic risk. *The American Journal of Clinical Nutrition.* 2013;99(3):454-462.
12. Eriksen R, Gibson R, Lamb K, et al. Nutrient profiling and adherence to components of the UK national dietary guidelines association with metabolic risk factors for CVD and diabetes: Airwave Health Monitoring Study. *British Journal of Nutrition.* 2018;119(6):695-705.
13. Pan X-F, Magliano DJ, Zheng M, et al. Seventeen-Year Associations between Diet Quality Defined by the Health Star Rating and Mortality in Australians: The Australian Diabetes, Obesity and Lifestyle Study (AusDiab). *Current Developments in Nutrition.* 2020;4(11).
14. Donat-Vargas C, Sandoval-Insausti H, Rey-García J, Ramón Banegas J, Rodríguez-Artalejo F, Guallar-Castillón P. Five-color Nutri-Score labeling and mortality risk in a nationwide, population-based cohort in Spain: the Study on Nutrition and Cardiovascular Risk in Spain (ENRICA). *Am J Clin Nutr.* 2021;113(5):1301-1311.
15. Mytton OT, Forouhi NG, Scarborough P, et al. Association between intake of less-healthy foods defined by the United Kingdom's nutrient profile model and cardiovascular disease: A population-based cohort study. *PLOS Medicine.* 2018;15(1):e1002484.

3. Besides demonstrating predictive validity of the nutrient profile model in relation to healthy dietary patterns and improved health outcomes. This paper also explored how each of the 9 scoring domains of the Food Compass Score separately related to dietary and health outcomes [Line 86] but the authors barely explore the implications of their findings here. This paper would be a much more interesting paper if it did explore some, at least, of those implications. Some commentary here (if not further analyses) in relation to the contribution of domain scores to final Food Compass cores is recommended.

The authors show, for example, that ‘the i.FCS for individuals was highly correlated with HEI-2015 (R=0.82) (Figure 2)’ but that: ‘Individually, the 9 domains of the FCS were not as highly correlated, ranging from 0.16 for i.Specific Lipids to 0.71 for i.Nutrient Ratios (Figure S2)’ [Line 268]. So doesn’t this mean that using the i.Nutrient Ratios domain for the i.FCS gives you almost all you need to give you a ‘good enough’ ranking of diets at least compared with the HEI 2015? What then is the added benefit of including, for example, the i.Specific Lipids domain in the total i.FCS and by extrapolation the Food Compass Score for foods and can this be quantified?

This matters because the authors continue, in my view, to fail to justify the inclusion of 54 components in the FCS when most other nutrient profile models manage with much fewer and also because the information needed to score many of the components of the Food Compass nutrient profile model is unavailable in many food composition databases other than that used by the authors in this paper. E.g. trans fatty acid information is commonly unavailable in many food composition databases and so too is the information necessary to score foods for degree of processing (here according to the NOVA classification system).

Similarly the authors show significant inverse associations with all-cause mortality for seven of the nine domains of the i. Food Compass Score [Line 329], i.e. for Nutrient Ratios, Vitamins, Additives Processing, Fiber and Protein, Specific Lipids and Phytochemicals and not for Minerals and Food Ingredients [Table S8]. Again does this not suggest that the Mineral and Food Ingredients domains could usefully be omitted from the final score?

Thank you for your suggestion to include a more thorough discussion around the i.Domain Scores. We agree that the ability to disaggregate Food Compass into these component domains, and assess their respective association with health outcomes, is an important feature of this analysis and NPS.

Importantly, the FCS domains were not selected or constructed based on the associations of these domains with healthy diet patterns or with health outcomes: that would represent a circular argument, where an NPS is over-fitted based on findings on health associations in the same dataset, and then termed “valid.” As described in our original paper, relevant attributes and scoring principles were developed based on assessment of more than 100 reported NPS, including 7 widely-used NPS of diverse origins; a systematic review of national and international dietary guidelines; nutrient requirements for health claims; and assessment of nutrients, ingredients, and other food characteristics linked to health outcomes. The domain system of Food Compass was also purposefully designed so that no one domain or attribute drives the overall product score—a mechanism to prevent industry reformulators from “gaming” the system.

Of the i.Domain Scores, i.Nutrient Ratios has the highest correlation with the HEI 2015. However, the correlation with HEI 2015 is just important parameter we assessed – the extent to which each i.Domain Score tracks with health risk factors, prevalent disease conditions, and mortality is also relevant.

The finding that the strongest association with HEI, and also the strongest associations with health outcomes overall, is for i.FCS, rather than any one domain, corroborates our development methods that assessing multiple domain scores in aggregate provides the strongest overall metric.

Finally, it's important to note that the primary purpose of Food Compass (and thus its associated domains) is not to derive an individual person's energy-weighted dietary score, and associate that with health outcomes. Rather, Food Compass was designed as a measure of the healthfulness of specific individual food and beverage products, and also mixed meals. When a consumer, business, or policy maker is making a decision about an individual product (e.g., to purchase, to reformulate, or to apply a positive or negative policy), a high degree of discrimination is required among different products. In our prior report, we assessed the face/content validity, convergent validity, and discriminatory validity of Food Compass across 8032 products representative of the US food supply. Food Compass demonstrated excellent content validity – the distribution of scores for major and sub-food categories as well as individual product scores were consistent with what would be expected for more healthful vs. less healthful foods. In addition, Food Compass was able to discriminate healthfulness within food categories, as well as within the NOVA classification for food processing. When compared to other commonly accepted NPS (Nutri-Score and Health Star Rating), Food Compass generally scored more consistently with the latest nutrition science – i.e., scoring refined grains products less favorably (lower scores) and scoring products with healthy fats (plant oils, seafood, etc.) more favorably than by Nutri-Score or Health Star Rating. Food Compass also does so using a single, consistent algorithm across all food and beverage types, rather than having differing, subjective scoring principles and thresholds, selected in a posthoc fashion, across different food and beverage categories. Thus, this present new paper is not aiming to reconstruct the Food Compass, which has shown previous validity as a measure of product healthfulness, based on associations of the energy-weighted i.FCS for persons with health outcomes, but to provide further validation. We have done so. The previous validation findings, coupled with the present predictive validation findings of significant associations with a healthy dietary pattern, a range of health risk factors and prevalent diseases, and total mortality support the validity of Food Compass as a measure of product healthfulness.

We have added further comments on these important points in the Discussion:

“The ability to score food and beverage items across 9 component domains is an important feature of the FCS and current investigation. Among the 9 domains, applied to individual items and then energy-weighted to persons' diets, none by itself had as strong an association with HEI, nor with the full range of health risk factors, prevalent diseases, and total mortality, as the overall i.FCS. At the same time, each i.Domain Score had some associations with HEI as well as varying strengths of associations with different health endpoints. These findings support the complementary nature of the different domains, each providing supportive and somewhat distinct information, as components of the Food Compass. The growing evidence for complex, heterogenous effects of diet on health (e.g., via the gut microbiota, epigenetics, etc.)⁴⁵ further supports the utility of a more holistic, multi-domain measure of healthfulness of foods and beverages. In addition, the domain structure ensures that no one domain or attributes can drive the overall product score—a mechanism which prevents mis-scoring based on extreme values of a single or few nutrients as well as industry “gaming” the system by fortifying food products with isolated

vitamins. This holistic, domain-based scoring also permits a single, consistent scoring algorithm across all food and beverage products, including mixed meals – in contrast to all other major current NPS, which require subjective grouping of foods and beverages into multiple categories that use differing algorithms and/or scoring thresholds and have trouble scoring mixed meals that contain ingredients across two or more of these categories.¹⁵”

Specific comments

Line 74. ‘The Food Compass is a novel NPS that incorporates a range of 54 protective and risk factor nutrients, ingredients, bioactives, additives and processing attributes, grouped across 9 domains, and selected and weighted based on the latest evidence about their relative healthfulness. The Food Compass Score (FCS), ranging from 1 (least healthful) to 100 (most healthful), enables a more consistent and universal algorithm across all food and beverage categories; permits similar scoring for mixed dishes and meals; and improves convergent and discriminatory validity of product scoring compared to other major NPS.

The comparative novelty of the Food Compass system does indeed mainly lie in the number and type of food attributes that are encompassed but in my view provides no more consistent scoring or convergent validity than other nutrient profile models. Furthermore I continue to fail to see why having more components than other nutrient profile models ‘enables a more consistent and universal algorithm across all food and beverage categories’ as the authors claim. But I have said this before in my review of the previous paper by Mozaffarian et al and the criticism seems to have gone unheeded.

The combination of its attributes and domains, together with scoring per 100 kcal, does allow Food Compass to have a consistent and universal algorithm, *per definition*, as the same algorithm is applied across all food, beverage, and meal product categories. Other NPS such as Nutri-Score and HSR require multiple separate scoring thresholds and algorithms for different food categories, to make the scores “make sense” based on what would be expected to be a healthy choice. We did not develop Nutri-Score or HSR, so we cannot say which aspects of these scoring systems required the subjective categorization of different foods and beverages into differing scoring algorithms and cutpoints. We can say that Food Compass, with its design, uses a single algorithm for all products as well as mixed dishes and meals.

Line 162 ‘To extend FCS for specific products to an individual’s overall diet, the scores for each item reported in a person’s diet were summed, weighted by its percent contribution to that person’s total energy intake and then used to calculate an individual’s Food Compass Score (i.FCS).’ There are many different ways of converting nutrient profile scores for foods into diet quality scores for individuals [e.g. see <https://discovery.ucl.ac.uk/id/eprint/1369569/1/Thesis-GMASSET-UCL-2012-FINAL.pdf>] It would be useful if the authors could explain their choice at greater length (e.g. as they say [Line 430] ‘Use of energy-weighting to calculate i.FCS provides lower weighting to certain foods with lower calories per servings such as fruits and vegetables. It seems to me to be likely that the particular way nutrient profile model scores should be converted to diet quality scores depends on the use to which the nutrient profile model is put (e.g. it will be different, say, if the model is for front-of-pack labelling purposes as opposed to restrictions on the marketing of foods to children)

We acknowledge that NPS can be converted to dietary scores for individuals in different ways. Energy (kcal) is the most natural unit to combine foods in a comparable way, e.g. comparing 100 kcal of

one product to another, for assessing health outcomes. Use of other metrics, like weight, can be highly problematic, due to bias from water weight, fiber, and fat. For example, for NPS that use grams, 100g of pork fat contains 638 kcal, while 100g of apple contains 52 kcal. Comparing the nutrients provided by these two foods using 100g, pork fat provides far more of many nutrients, leading to absurd claims that pork fat is the 7th healthiest food in the world (<https://www.bbc.com/future/article/20180126-the-100-most-nutritious-foods>, <https://journals.plos.org/plosone/article?id=10.1371/journal.pone.0118697>). Because the original Food Compass Score for products is based on kcal, combining these product scores based on kcal is most appropriate for generating a person's dietary score.

We have added a sentence to the Discussion:

“Both the FCS and i.FCS were weighted by energy (kcal), rather than weight or portion size, which prevents bias from differences in water weight, fiber, or fat content. For example, in a recent NPS that evaluated foods per 100g, pork fat was identified as the 7th most nutritious food in the world,^{56,57} ignoring the obvious problem that 100g of pork fat (and its nutrients) contains 638 kcal, compared for example to 100g of apple which contains only 52 kcal (due to higher water content), creating a flawed comparison of the total nutrients provided by each 100g portion.”

Line 416 ‘in contrast, several validation studies of other NPS (40-43) utilized food frequency questionnaires that only allow scoring of major food categories.’ This is a selective comparison. There are many other validation studies of nutrient profile models that have used dietary data from multiple 24-hour recalls (e.g. validation studies of the NutriScore algorithm involving the NutriNet Sante cohort and the SUVIMAX study.

(e.g. https://academic.oup.com/eurpub/article/30/Supplement_5/ckaa166.1285/5914738?login=true

To be clear, we were not claiming that all prior NPS validation studies utilized FFQs, but that “several” did. We have clarified the statement:

“While some prior NPS validation studies utilized detailed product information from 24 hour recalls,⁴⁶⁻⁵¹ several utilized food frequency questionnaires⁵²⁻⁵⁵...”

Reviewer #2 (Remarks to the Author):

This paper follows on from a publication by the same group in Nature Foods in 2021 which described a new nutrient profiling score (Food Compass). This tool (like other NPSs) is aimed at assessing the healthfulness of individual foods. This perspective of individual foods links this NPS (and others) to the issue of front of pack labelling.

The purpose of this new paper is different as it aims to take the Food Compass score for each food eaten by an individual and then to summate it to provide a summary overall score. This is intended to be an indication of the healthfulness of the diet than individual foods. What the authors show is in an analysis of NHANES that this score relates to various risk factors and prevalent disease in a way that one might expect and that it is predictive of overall mortality in a follow up analysis over nearly 17 years in which there were nearly 5000 deaths in the cohort of 38K individuals included in the prospective analysis.

This analysis does suggest that the individual Food Compass Score does have predictive validity for total mortality in a US population.

There are some minor points as below about the analysis but the major question is not about whether the iFCS has predictive validity or not but rather whether that has any utility either for public health, for clinical practice or for the people themselves.

The authors state that "NPS like Food Compass can be one important tool in such interventions"... (to improve diet quality)... "allowing consumers, industry including (food manufacturers, retailers, and restaurants,) investors, schools, hospitals, and worksites, and policy makers to identify and shift toward healthier food and beverage options". Whilst one can understand how front of pack labelling using some form of NPS might influence individual food choices, it is rather difficult to see how the computation of an overall dietary score using the iFCS would have an impact. The argument for how computation of the iFCS might actually be used, by whom and for what purpose, needs to be much clearer. It is not reasonable to merely state the argument for the individual food NPS because that is not what this paper is about. A clearer explanation of the usefulness of an overall diet NPS is required.

Thank you for these comments. We agree with these points. The primary purpose of the i.FCS was not as a new metric for public health or clinical practice, but as a validation tool of the FCS for assessing individual products.

Previous analyses demonstrated that Food Compass has good face validity, convergent validity, and discriminatory validity for individual food and beverage products (Mozaffarian *et al.* 2021), which could be used for consumer guidance, policymaking, and product reformulation. To further confirm the validity of FCS, we developed i.FCS for predictive validation of construct validity – the association of the scoring system with health outcomes, when applied to a consumer's choices for diet.

We are not proposing that i.FCS be used for other purposes.

The basis for the calculation of the iFCS is both somewhat arbitrary and very complicated. Why the different domains are equally weighted with the exception of three which receive 0. weights is not explained. Why only the top 5 vitamins and minerals and the top 3 specific lipids are included is not self-evident.

The iFCS is calculated directly from the product scores, which are then weighted by the energy content of each product to derive the iFCS. The i.Domain scores of individuals are not separately used to derive the iFCS. Rather, they were separately calculated to explore their associations with the endpoints of this study.

We have clarified in the Methods:

“The i.Domain scores were not used to derive the i.FCS, but separately calculated from the product-level domain scores to explore their associations with the endpoints of this investigation.”

The correlation with the Healthy Eating Index is 0.82. This does beg the question about whether the iFCS has any advantage over the HEI. It is certainly not easier to compute. Some discussion of their relative merits is warranted.

Please see the comments above. We agree. We are not aiming to replace HEI, but using HEI as a “gold standard” dietary pattern to validate the utility of the FCS for scoring individual food and beverage products. (HEI, of course, cannot score specific food and beverage products)

The iFCS quintiles appear to be strongly inversely related to total energy intake and there is a weighting for each food stuff on the basis of their contribution to total energy and it appears at least in some of the analyses that TEI was also included as a covariate. A clear statement about what role TEI plays in this score would be helpful.

This is an excellent point. TEI is not included as a component of either the FCS or the iFCS. Thus, adjusting for TEI as a potential confounder is important. We have clarified this in the Methods text:

“Average total energy intake was not included as a component of either the FCS or the i.FCS. Thus, we derived total energy intake from the 24-hour recall data available and adjusted for it continuously as a potential confounder.”

There is a lot of missing data in this study. Some variables e.g fasting glucose, LDL-cholesterol and triglycerides are missing in more than 50% of people. In such a situation of extreme missingness, imputation with whatever method is problematic. There are also some challenges with physical activity which was assessed differently in different time periods. In this case one can understand how the use of the different measures of PA can be used to impute missing values. It is less clear how nutrients that are not assessed at all (as far as I understand it) e.g total flavenoids, vitamin D and choline can be estimated by imputation.

In NHANES, certain fasting laboratory measures are missing in a significant proportion of participants because the participants were not examined in the fasting state. We included complementary biomarkers of long-term glucose homeostasis (HbA1c) and blood lipid profiles (HDL-C, total cholesterol: HDL ratio) with less missingness; and these findings were generally consistent. We have clarified these points in the Discussion limitations:

“Certain fasting biomarker levels (e.g., triglycerides, fasting plasma glucose) were missing for a significant proportion of the NHANES sample, which may have weakened the accuracy of the estimates

for these health indicators. However, assessment of complementary biomarkers with less missingness for long-term glucose homeostasis (HbA1c) and blood lipid profiles (HDL-C, total cholesterol: HDL ratio) yielded generally consistent findings.”

The methods to assess physical activity changes over time in NHANES from 1999 to 2018. As detailed in the Supplemental Materials, we utilized imputation so that the older PA measures could be more “standardized” to the newer methods, allowing PA to be used as a covariate in the analyses.

Nutritional composition for all foods in NHANES is reported in the USDA’s Food and Nutrient Database for Dietary Studies (FNDDS). Because we utilized multiple NHANES cycles, a few nutrients were missing in certain cycles. For example, FNDDS reporting on vitamin D and choline started in 2007-2008 and 2005-2006, respectively. To account for missingness in these attributes for the earlier survey cycles, we first carried values backwards for the same products reported in different cycles; and then used multivariable imputation to estimate the remaining missing values. Similarly, the USDA has a database of total flavonoid values per 100g for reported food and beverage products from 2007-2010. We used total flavonoid values available from this database first, carried forward and backward for the same products reported in different cycles; and then used multivariable imputation to estimate the remaining missing values. Any errors in nutrient imputation would unlikely be systematic with respect to disease outcomes or total mortality, and so this imputation is likely to attenuate findings toward the null. We have added this to the Discussion limitations:

“Some nutrients such as Vitamin D, choline, and flavonoids were only available in certain NHANES cycles, requiring imputation in other cycles. Because errors in nutrient imputation would unlikely be systematic with respect to disease outcomes or total mortality, this imputation may have attenuated findings toward the null.”

There is an important typo in the description of the means which states that the mean HbA1c in the population is 6%. I think this should read 5.6%.

We have corrected this to 5.6%, thank you

Reviewer #3 (Remarks to the Author):

This manuscript describes an elaborate food scoring system, which has been extended to assessment of an individual's dietary intake. Such scoring ensures like-with-like comparisons and may therefore be used to estimate food quality at point of purchasing as well as diet as a whole. Here, the authors validate the score by cross-sectional as well as prospective indicators of (ill) health. These associations were observed to be largely significant though relatively small in effect size and have plausible biological mechanisms with the domains of the food compass score.

The algorithm behind the food compass is extensive and "not for the faint-hearted". It captures a large range of dietary exposures which otherwise are covered in separate scores. An extensive and elaborate work which emphasizes the complexity and inter-relationships of foods which make up our diet.

Thank you for your interest in our analysis and detailed critical feedback on both the manuscript and supplement. We appreciate your attention to detail and important perspectives.

My comments are divided in two parts. The first part relates to methodology/interpretation, the second part are just suggestions for the text.

Introduction

Line 51: Please include references after the first sentence.

We have added four references now to support this sentence:

1. El-Abadi, NH, Taylor, SF, Micha, R, & Blumberg, JB. Nutrient Profiling Systems, Front of Pack Labeling, and Consumer Behavior. *Current Atherosclerosis Reports*. 2020; 22(8): 36. doi:10.1007/s11883-020-00857-5
2. Labonté, M, Poon, T, Gladanac, B, et al. Nutrient Profile Models with Applications in Government-Led Nutrition Policies Aimed at Health Promotion and Noncommunicable Disease Prevention: A Systematic Review. *Adv Nutr*. 2018; 9(6): 741-788. doi:10.1093/advances/nmy045
3. McColl, K, Lobstein, T, Brinsden, H, & World Health Organization. Regional Office for, E. Nutrient profiling could be used to transform food systems and support health-promoting food policies. *Public health panorama*. 2017; 03(04): 586-597
4. O'Hearn, M, Gerber, S, Cruz, SM, & Mozaffarian, D. The time is ripe for ESG + Nutrition: evidence-based nutrition metrics for Environmental, Social, and Governance (ESG) investing. *Eur J Clin Nutr*. 2022. doi:10.1038/s41430-022-01075-9

Line 55: "industry reformulation" or is meant "food reformulation by the industry"?

Good point. We have clarified this in the text that it is "food reformulation by industry".

Methods

Line 93: Please include the upper age range.

The upper age limit is 85 years. We've updated the methods text accordingly.

Line 109: Which criteria were applied to assess completeness and validity of the diet record data?

We used the NHANES “dietary recall status code” (DRDDRSTS [1990-2000 only]; DR1DSTZ [Day1] or DR2DSTZ [Day2] for 2001-2018) in the dietary intake files, an indicator of the quality and completeness of a survey participant’s response to the dietary recall section. We excluded individuals with code 2, 3 or 5. We excluded individuals < 20 years, which covered code 4 (reported consuming breastmilk). Code details in the table below:

Code	Code name	1990-2000	2001-2 – 2017-18 survey cycles
1	Reliable and met minimum criteria	Record was verified as reliable and met the following minimum criteria: A. Less than 25% foods with missing descriptive information (e.g., caffeinated or decaffeinated, preparation methods, or brand names); B. Less than 15% foods with missing amounts; and C. Any meal reported must have at least one known food. For example, if a respondent reported having a lunch but could not remember any foods from that lunch, the recall did not meet the criterion.	Record was verified as reliable and met the following minimum criteria: A. The first 4 steps of the 5-step AMPM completed. B. Food/beverages consumed for each reported eating occasion identified.
2	Reliable but did not meet the minimum criteria	Dietary recall code was verified as reliable but did not meet minimum criteria as stated in 1	No data on total nutrient intakes and the total number of foods reported are provided for these cases
3	Not reliable	The dietary recall information provided by the respondent was determined to be unreliable. No data on individual food consumption or total nutrient intake was provided in the dataset for unreliable cases.	Code not included in 2001-2002 survey cycle onwards
4	Reported Consuming breastmilk / Breastfed infant or child	The foods reported during the dietary recall interview included human milk. Few respondents could quantify the human milk intake for their breast-fed infants/children. The foods consumed by nursing infants and children are reported in the Individual Foods File. For those who could not quantify the amount of human milk intake, gram weight consumed and nutrient values of the human milk records were set to missing.	
5	Not done	The dietary recall section of the interview did not take place due to various reasons (e.g. came late/left early, refusal, illness, emergency, or equipment failure).	

We’ve updated the Supplementary Materials (new Text S1) to clarify the criteria used:

“We used all complete and reliable dietary records available in the NHANES 24 dietary assessment,⁶ as determined by the National Center for Health statistics staff, averaging the two recalls for each individual when present to reduce the influence of day-to-day dietary variation.

Reliable 24-hour dietary recalls were defined as meeting the following criteria in 1990-2000 survey cycle:

- *Less than 25% foods with missing descriptive information [e.g., caffeinated or decaffeinated, preparation methods, or brand names]*
- *Less than 15% foods with missing amounts, and any meal reported must have at least one known food;*

And the following updated criteria for the 2001-2 to 217-18 survey cycles:

First four steps of the 5-step USDA automated multiple-pass method (AMPM) completed and food/beverages consumed for each reported eating occasion⁶”

Line 116: Why are alcoholic beverages excluded from the score?

While alcoholic beverages are sources of calories and nutrients for many individuals, these products effect health through alternative, non-nutritive pathways, outside the scope of Food Compass. Mozaffarian *et al.* 2021 describes the exclusions of alcoholic beverages, among other products, from Food Compass scoring as well. Given the known associations of excess alcohol consumption and poor health, we included alcohol intake (as a percentage of energy) as a covariate in our analyses.

Line 149: The manuscript mentions 54 attributes, but Table S1 indicates that certain attributes are not used in the score due to unavailability of data in the various composition tables. Trans fats are marked as not included, but appear in the various tables. Text S1 may also need checking for similar reasons.

The Food Compass algorithm ideally incorporates 54 attributes. However, due to lack of data in FNDDS, certain attributes were not available. We have updated the Text S1 to mention the exclusion of trans fats. We have also added the following sentence to the methods text: *“In addition, due to data availability in FNDDS, 7 attributes were excluded from Food Compass scoring (Table S1).”*

Line 158: It would be good to know more about the motivation for these cut-off values (<30, 30-70, >70), what the reasoning behind these has been. Also, when taking this into a public health setting, it would be good to see how the frequency distribution of these three categories is in relation to the i.FCS consumed. In other words, is it about consuming more >70 foods, less <30 foods, or the relative proportion between them?

As described in our prior report, Food Compass was applied to 8032 unique foods and beverages from the USDA’s Food and Nutrient Database for Dietary Studies. The observed distribution of Food Compass Scores was assessed overall, across 12 major and 44 minor food categories. The 25th and 75th percentiles of the overall food scores were about 30 and 70, respectively. On the basis of the observed ranges overall and within food categories, FCS \geq 70 was selected as a reasonable cut-point for foods and beverages to be encouraged; FCS = 31-69, to be consumed in moderation; and FCS \leq 30, to be minimized. We have updated the text to clarify that these cut-off values are “based on the observed ranges of the FCS distribution of scored products[cite our prior paper]”.

Your question about the frequency distribution of these 3 categories in relation to i.FCS is excellent, and one that we had not previously considered. We have performed additional analyses of the counts of different products with FCS \geq 70; FCS 31-69, and FCS \leq 30 in each of the three categories of the i.FCS score (\leq 30, 31-69, and \geq 70). We have also created a visualization of 12 sample NHANES participants with i.FCS scores between 40-48, and the respective %E and num of foods from each FCS category (FCS \geq 70; FCS 31-69, and FCS \leq 30) to convey the heterogeneity in dietary composition that lead to a given i.FCS diet score. These methods and findings have been added to the Manuscript -see table S5, Figure 2 and additional manuscript text below.

Table S5. Number and contribution to energy of specific foods and beverage products (FCS \geq 70; FCS 31-69, and FCS \leq 30) among individual people with i.FCS \leq 30, 31-69, and \geq 70.*

i.FCS for individual persons		
\leq 30	31-69	\geq 70
(n = 15,714 individuals, 32.7%)	(n=32,069 individuals, 66.8%)	(n = 216 individuals, 0.5%)

Contribution of food and beverage products in each FCS category †

Products with FCS ≤30			
Count (median, IQQ)	11 (7, 5)	8 (5, 12)	2 (1, 4)
% energy contribution (median, IQR)	65.0 (56.1, 74.4)	39.1 (28.3, 48.8)	5.9 (2.1, 10.4)
Products with FCS 31-69			
Count (median, IQQ)	5 (3, 8)	8 (5, 11)	5 (3, 8)
% energy contribution (median, IQR)	30.6 (20.8, 40.4)	44.7 (33.5, 56.0)	27.8 (18.7, 37.2)
Products with FCS ≥70			
Count (median, IQQ)	2 (1, 4)	6 (3, 9)	13 (8, 20)
% energy contribution (median, IQR)	2.3 (0.2, 6.1)	14.2 (7.3, 23.2)	65.5 (57.9, 71.4)
Total number of products reported	20	24	24

* The individual Food Compass Score (i.FCS) was calculated as the energy-weighted mean FCS of all foods and beverages consumed, as reported in 24 hour dietary recall. The overall score could range from 1 to 100.

†The FCS for each food and beverage was calculated ...

i.FCS, individual Food Compass score; IQR, interquartile range; FCS, Food Compass Score

Figure 2. Examples of individuals having the median U.S. i.FCS score of 35.5, demonstrating variation in the consumption of products with FCS ≤30, 31-69, and ≥70 that achieves a similar overall score. The distribution (count and contribution to total energy intake [%]) of consumed food and beverage products, based on previously defined product healthfulness thresholds (i.e., FCS≤30 as products to minimize; FCS 31-69 as products to be consumed in moderation; and FCS ≥70 as products to be encouraged) was assessed for all NHANES participants. Examples are shown for 10 participants with mean i.FCS at the U.S. median (35.5 ±1) and consuming at least 10 different products over 2 days of diet recalls. The numbers within each stacked bar graph indicate the count of food and beverage products consumed from that category of FCS score across two days of reported intake; and the color bars represent the percentage energy contribution of food and beverage products from that category of FCS score across two days of reported intake.

HS, high school; grad, graduate

Methods text added:

“To understand how different combinations of items could contribute to a similar i.FCS, we assessed the distribution (number and contribution to total energy intake) of consumed food and beverage products,

based on previously defined product healthfulness thresholds of $FCS \leq 30$, 31-69, and $FCS \geq 70$, among subgroups of the US populations having various i.FCS scores.”

Results text added:

“Individuals with higher i.FCS scores (≥ 70) consumed a greater number of and percentage total energy contribution from products with $FCS \geq 70$ (median [IQR] count: 13 [8, 20]; percentage energy: 65.6% [57.9, 71.4%]) compared to products with FCS 31-69 (6 [3, 9]; 14.2% [7.3, 23.2]) or $FCS \leq 30$ (2 [1, 4]; 2.3% [0.2, 6.1]) (Table S5). In comparison, individuals with lower i.FCS scores (≤ 30) consumed a greater number of and percentage energy contribution from products with $FCS \leq 30$ (11 [7, 15]; 65.0% [56.1, 74.4]) compared to products with FCS 31-69 (8 [5, 12]; 39.1% [28.3, 48.8]) or $FCS \geq 70$ (2 [1,4]; 5.9% [2.1, 10.4]). At the same time, however, for any i.FCS score, there was also substantial heterogeneity in the counts and energy contribution from products of different FCS scores (Figure 2), indicating that different people could arrive at similar overall i.FCS scores in different ways.”

Discussion text added:

“Notably, different people could arrive at a similar i.FCS scores with very different combinations of healthier or less healthy foods, and yet the overall i.FCS was still predictive of a healthy diet pattern, health risk factors, prevalent disease conditions, and total mortality.”

Line 166: Please clarify that the individual domain scores also range from 0-100.

We have updated the text to specify that the individual domain scores also were standardized to a potential range from 1-100 (similar to FCS and i.FCS).

Line 203: Considering the HEI and the FCS have the same range, apart from showing the scatterplot and correlation between the two scores (figure 2), it would be nice to see the agreement in the form of a Bland-Altman plot.

This is an interesting suggestion. Bland-Altman plots are designed to identify systematic differences for two instruments measuring the same thing, i.e. repeated blood pressure measures using two different cuffs. In our analysis, we are not comparing two assays designed to measure the same thing, but comparing i.FCS to a gold standard dietary pattern measure for validation purposes. Thus, we don't think it is appropriate to include this in the analysis.

Results

Line 285: It would be good to have Table S7 in the main text, since it shows the associations for the separate domains and emphasizes their relative importance for the presented risk factors/conditions. Did the results in S7 change much after adjustment for daily energy intake? Considering the domain “additives”, are the observed associations mainly driven by the attribute “added sugar”? Also, protein and fiber are positively associated with prevalent diabetes, may this be due to reverse causality?

This is a good suggestion. Given Nature Communications Article instructions allowing for up to 10 display items, we have moved the Table S7 to the main manuscript, and it is now Table 3. We found adjusting for energy intake and survey cycle did not appreciably alter the observed associations with cross-sectional health biomarkers and prevalent conditions.

Regarding the i.Additives domain: “added sugar” and “nitrites” attributes contribute equally to the i.Additives Domain score. However, given the US food supply has more foods with non-zero added sugar values than non-zero nitrite values, it is reasonable to assume that i.Additives domain scores for many Americans are driven more by intake of foods with added sugar.

To further explore the association between the i.Fiber and Protein domain and prevalent diabetes, we conducted sub-analyses disaggregating the Fiber and Protein attributes and found that Fiber was associated with lower odds of prevalent diabetes, while Protein was associated with higher odds of prevalent diabetes. In meta-analyses of diverse long-term prospective cohort studies, higher protein intake is associated with higher incidence of diabetes; and in randomized trials, higher protein intake drives hepatic de novo lipogenesis (the conversion of excess dietary carbohydrate and protein energy into fat for long-term storage), which is one of the driving pathways for fatty liver, visceral fat accumulation, and insulin resistance. Further interventional studies are required to assess the causal mechanisms driving the association between foods and diets higher in protein intake and increased diabetes risk. We have added these exploratory analyses to the Results, and highlighted these points in the Discussion.

Added results text:

“In further post-hoc exploratory analyses disaggregating the i.Fiber and Protein domain into its individual attributes (i.Fiber and i.Protein), i.Fiber was associated with lower prevalence of diabetes, while i.Protein was associated with higher prevalence of diabetes (data not shown).”

Added discussion text:

“The observed harmful association between the i.Protein attribute and prevalent diabetes is consistent with meta-analyses of prospective cohort studies which identified a positive association between higher protein intake and higher incidence diabetes³⁴; as well as randomized trials where higher protein intake was associated with hepatic de novo lipogenesis³⁵ – a driving pathway for fatty liver, visceral fat accumulation, and insulin resistance.”

Line 274/294/313: Considering the wide age range (20-85+) in the sample, do the authors consider the associations between the i.FCS and various outcomes to be the same across the whole sample?

This is an excellent point. We have now explored the potential for heterogeneity in the association between i.FCS and total mortality by age, as well as by sex, education level, race/ethnicity, and income level. We have included these new analyses in a new Table S10, as well as the Methods and Results.

Table S10. Multivariable adjusted, survey-weighted prospective association between individual Food Compass Score (i.FCS) and all-cause mortality, stratified by key sociodemographic factors, among U.S. adults, 1990-2018

Sociodemographic factor	N	Events (No. of deaths)	i.FCS, per SD (10.9 points) *
			HR (95% CI)†
Sex			
Male	23015	4164	0.93 (0.88, 0.98)
Female	24937	3328	0.93 (0.88, 0.98)

Age category, years			
20-64	36098	2225	0.90 (0.84, 0.96)
≥65	11854	5267	0.94 (0.89, 0.98)
Race/ethnicity‡			
Hispanic American	12320	1272	0.88 (0.78, 0.98)
Non-Hispanic White	21581	4518	0.93 (0.89, 0.98)
Non-Hispanic Black	9948	1465	0.95 (0.89, 1.02)
Education level			
≤HS graduate	23906	4841	0.94 (0.89, 0.99)
>HS graduate	24046	2651	0.90 (0.85, 0.95)
Income, PIR§			
Low income (PIR ≤2.16)	24104	4374	0.91 (0.86, 0.96)
High income (PIR >2.16)	23848	3118	0.93 (0.88, 0.98)

* Individual Food Compass Score (i.FCS) calculated as the energy-weighted mean of FCS of all foods consumed, as reported in up to two 24 hour dietary recalls per person, with a potential range from 1 to 100.

† Survey-weighted, multivariable-adjusted Cox proportional hazard model incorporated NHANES dietary recall sample weights to account for the complex survey design and response rates and provide nationally representative effect estimates and associated 95% confidence interval for the non-institutionalized U.S. population. All models adjusted for: age (years), age² (years), sex, race/ethnicity (Mexican-American, other Hispanic, non-Hispanic White, non-Hispanic Black, Asian/other race), education level (<HS graduate, HS graduate, some college or associates degree, ≥ college graduate), income (poverty: income ratio) unless stratifying for that variable; and smoking status (non-smoker, former smoker, current smoker); total physical activity (MET-hours/week), alcohol use (%energy), and self-reported diabetes (yes, no).

‡ Mexican-American and Other Hispanic combined, and Asian/other excluded, from stratified analysis
AA, Associates Degree; CI, confidence interval; HS, high school; i.FCS, individual, dietary Food Compass Score; PIR, poverty: income ratio; SD, standard deviation

§ Represents the ratio of family income to the federal poverty threshold, adjusted for household size. A higher ratio indicates a higher level of income. Individuals were grouped into low and high income, with the cut-point drawn at the median PIR (2.160).

We added the following sentence to the method text:

"In exploratory analyses, we investigated the relationship between i.FCS and total mortality in subgroups by age, sex, race/ethnicity, education, and income to assess potential variation (interaction) in the association according to these key sociodemographic factors."

We also added the following paragraph to the results text:

"Findings were similar across subgroups, with no significant differences in the observed protective associations between i.FCS and mortality (Table S10)."

Figures

Figure 1: What is the reason for the 'unevenness' in the distribution for additives?

The distribution of the i.Additives domain indicates that the energy contribution of the majority of Americans' diets was not from foods with high amounts of nitrites or added sugar. First, nitrite-containing foods (i.e. processed meats) are a relatively small proportion of the total US food supply;

and there is also little overlap between foods with both high nitrite and high added sugar content, so getting a very low score would not be common.

Figure 2: The Y-axis ends on 00, but this is presumably 100.

Thank you, we have now fixed this figure.

Figure S3: This figure is not referred to in the text. It models the domains individually in their association to all-cause mortality (mutually adjusted for). How are these positive and negative associations interpreted/explained?

In this exploratory analyses, the associations can be interpreted as the risk of mortality given a 1 standard deviation change in each i.Domain Score, after mutually adjusting for other i.Domain Scores, demographics, and lifestyle factors. It suggests that each domain is generally insufficient alone, when considering the contributions of the other domains, to predict mortality risk. These domains were not designed nor intended to be considered adjusted for each other, but to be added together synergistically. We have removed this exploratory analyses from the paper.

Tables

On various occasions in the S-tables, the zero (?) behind the decimal point is missing.

We have manually updated this output error.

Table 2: Could total cholesterol be added?

Yes, but as you know, total cholesterol alone is just a screening biomarker; and the clinically predictive and utilized metrics in practice are LDL-C, HDL-C, triglycerides, and the total cholesterol: HDL ratio. Thus, we have elected not to add TC as another blood lipid profile markers.

Table S1: Could an additional column (as first column) be added which indicates 'included' or 'excluded' from the FCS, thereby shifting any attributes listed but not included in the i.FCS into the row for 'excluded'.

This is a helpful comment. We have now included a footnote (§) which specifies which attributes were excluded from this analysis due to data unavailability in FNDDS.

Table S2: What is meant by 'total' behind some of the nutrients? Is this food and supplement sources combined? Or might this be e.g. a-TE instead of mg tocopherol. The footnote describes 'original score', what is meant by this?

'Total' in Table S2 refers to all food sources of that particularly vitamin. In the FNDDS database we used for this analysis, total folate (DFE) is reported as well as the disaggregated food folate and folic acid. For vitamin B12 and vitamin E, added vitamin B12 and alpha-tocopherol are also reported. We have now removed the "(total)" behind these micronutrients in the Table S2 to avoid confusion.

The first footnote refers to "original score" in the calculation of the final FCS. We've since clarified this to be the "unscaled score" that is calculated from the algorithm, before scaling from 1-100.

Table S6: adjustment for daily energy intake and survey cycle minimally changed the associations observed, apart from triglycerides and LDL-C. Just curious whether there is an explanation.

An interesting point. While the observed associations for triglycerides and LDC-C changed more when the models were adjusted for energy intake and survey cycle, the change was still within the confidence interval of the primary analysis – indicating that the difference in these two models was not statistically significant. We cannot think of a biological rationale for this change, so likely this (nonsignificant) change in the estimate is just a chance finding.

Discussion

Line 406: I agree this is a big advantage. However, the ability of the FCS to capture the various settings in which people eat (home, take-away, restaurant etc) requires a dietary assessment method (and extensive food list) which equally needs to be able to capture and differentiate on these aspects.

For assessing an individual's diet, this is true: 24-hour recalls will be needed. However, the purpose of FCS is not to assess an individual's diet, but to assess the healthfulness of products being sold to consumers. Thus, FCS can be used to score meals in cafeteria menus, restaurants, etc., which other major NPS cannot do. We have edited our discussion to reflect this:

"In this national dataset, Food Compass was able to score not only manufactured products but a person's entire diet, including complex home-cooked, cafeteria, and restaurant mixed meals which other major NPS generally cannot do."

Line 417: Even though the foods in the food composition table can be precisely categorized, the i.FCS may still suffer from systematic error due to misreporting and omission of food items by participants. Could the authors refer to the implications for this in the section on limitations?

This is a good point. If such error is random with respect to the outcomes, this will attenuate findings toward the null. If systematic, it will bias results in unpredictable directions. We have added these points to the limitations section:

"Misreporting and omission of food items by dietary recall participants was possible. If such error is random with respect to the outcomes, this would attenuate findings toward the null. If systematic, it could bias results in unpredictable directions."

Line 420: A very good point to make!

Thank you. We believe NPS validation studies that used FFQs for their dietary exposure data are limited in being able to validate the association between that NPS and health outcomes.

*Food composition tables may not contain all the details required to calculate the FCS/i.FCS. How do the authors see the extension of their work to other countries/settings? What would be their advice if certain attributes are missing?

We believe the primary advantages of the FCS are not the number of attributes, but other, more fundamental novel design features. These include: (a) the use of domains, which provides a more holistic assessment of foods and beverages while also preventing excess weight from any single attribute; (b) the integration of cutting edge-science in selection of the basic attributes, including the

use of nutrient ratios (unsaturated:saturated fat, potassium:sodium, fiber:carbohydrate) which more accurately capture fat, mineral, and carbohydrate quality, as well as the omission of outdated attributes which are major components of other NPS, in particular total fat and total calories; (c) the scoring on the basis of 100 kcal, rather than 100 g as in many other NPS; and (d) the incorporation of some features of processing, beyond nutrients alone.

For example, among the various domains, the most predictive was often the nutrient ratio domain. In addition, other analyses show that scoring by weight or serving size, rather than the natural unit of kcal, can be highly problematic, due to bias from water weight, fiber, and fat. For example, in a recent published NPS that scored foods based on 100 g, pork fat was declared the 7th most nutritious food in the world. This is entirely due to the bias of scoring by weight: 100g of pork fat contains 638 kcal, while 100g of apple contains 52 kcal. Comparing the nutrients provided by weight of these two foods, pork fat of course provides more of many nutrients per 100g given that it contains no water, leading to absurd claims that pork fat is the 7th healthiest food in the world (<https://www.bbc.com/future/article/20180126-the-100-most-nutritious-foods>, <https://journals.plos.org/plosone/article?id=10.1371/journal.pone.0118697>).

Thus, a more limited FCS, retaining these basic design strengths but employing fewer nutrients, will likely still be a reasonable predictor of the healthfulness of products. We plan to test this question in future manuscripts. In addition, imputation can be used to deal with missingness of certain variables – we are currently working on such imputation to apply FCS to hundreds of thousands of branded food products.

We have updated the discussion text with these important points:

" The Food Compass could be a possible unified standard NPS for these efforts, given its association with a healthy diet pattern and multiple health endpoints and its use of a single, consistent algorithm for all foods, beverages, mixed ingredients, and mixed meals. Such a standard may also provide a more accurate assessment of healthfulness for consumers, industry, and procurement decisions, in comparison to strategies using isolated nutrients or ingredients, such as the FOP warning labels recently implemented in Chile, Mexico, Uruguay, and Israel.⁴⁰ While FCS has more attributes than existing NPS, its core design strengths are the use of domains, which provides a holistic assessment of foods and beverages while also preventing excess weight from any single attribute; integration of cutting edge science in selection of predictive attributes (i.e., nutrient ratios which more accurately capture fat, mineral and carbohydrate quality) and omission of outdated attributes such as total fat or total calories; scoring on a basis of 100 kcal rather than 100 g, which greatly reduces bias from water weight; and incorporation of novel relevant features such as processing and additives. In future work, we plan to test FCS versions that retain these core design strengths while employing a more limited set of attributes; as well as leverage multivariable imputation to handle missing values in certain attributes in large datasets of branded products. Extension to datasets in low and middle income countries will also be important."

*For the nutrient scores, 25% of the RDAs for men 19-50 years were used, overestimating the domains for certain age-sex groups (e.g. iron, calcium). For intention of use as FoP labelling, this systematic error seems preferable to detail and clutter, but as i.FCS, would a more 'individualized' score taking age and sex into account when it comes to RDA be preferred perhaps? Have alternative algorithms been tried?

This is an excellent point. For practical goals of creating a single score, some assumptions and simplifications were needed. In the future, more "individualized" NPS could be crafted based on specific characteristics of the individual, such as age, sex, disease status, and more. We have added this point to the discussion:

“For practical goals of creating a single NPS score, some simplifications were needed, such as using the RDA for 19-50 year-old men as target high scores for several nutrients in the Food Compass algorithm. In the future, more “personalized” NPS could be crafted based on specific characteristics of the individual, such as age, sex, disease status, and more.”

*The impact of food production and consumption on the environment is becoming increasingly important. A process has been started for food guidelines to include aspects of sustainability. Could this be the 10th domain? Or would sustainability (like alcohol) be best modelled ‘outside’ the FCS?

We fully agree. Indeed, this was the original vision of the Food Compass, in that we anticipated each direction of the compass scoring a different feature food: e.g., healthfulness, sustainability, social justice, animal welfare. We plan in future work to add such additional independent dimensions to the Food Compass. We have added these points to the Discussion:

“Finally, the long-term vision of Food Compass is to scoring additional features of foods and beverages, such as environmental sustainability, social justice, and animal welfare – one for each direction of the compass. Future work is required to explore, add, and validate these additional dimensions.”

References

Please include an ‘accessed date’ for the web-based sources. **Done**

Ref 44: Could the authors include the chapter or possibly even refer to the original publication(s)?
Yes, we’ve updated this reference to include the chapter number and title from Willett’s Nutritional Epidemiology.

Textual

Line 25: Suggest removing “consistent”, since this is implied by “algorithm”.

We use the word consistent to refer to the fact that we use the same algorithm across all food and beverage products, rather than different algorithms for different categories of foods, as done with many other nutrient profiling systems like Nutri-Score, Health Star Rating, etc. We’ve updated the text to say “same” instead of “consistent” for greater clarity.

Line 34: Suggest adding “In cross-sectional analysis, [after multivariable ...]” **Done**

Line 35: Suggest to remove “levels of” in front of BMI. **Done**

Line 38: For better readability add “as well as, [lower prevalence]” **Done**

Line 73: Suggest rephrasing to “health risk related nutrients, ...”. We have rephrased it to: *“54 potentially protective and harmful nutrients,”* as some attributes are protective

Line 80: Suggest rephrasing to “...and the corresponding validity for the association between a person’s FCS and health outcomes, ...”. We updated it to read: *“and the corresponding validity for the association between a person’s FCS with a validated healthy dietary pattern and major health outcomes, have not been established.”*

Line 98: Suggest to include “...(NDI) up to 2015 we ...”. We’ve updated this now as NDI data is available through 2018

Line 117: Suggest including “Any missing attributes (i.e. nutrient quantities) required ...” **Done**

Consider to move the FCS paragraph after the dietary assessment, followed by the paragraph on socio-demographics. Good suggestion, we have moved the FCS paragraph after dietary assessment, followed by sociodemographic factors and covariates.

Line 162: Suggest to write the sentence in the order of actions “..., the FCS score for each food item reported in a person’s diet was weighted by its percent ...intake and then summed.”

We’ve updated the text per your suggestion to be more clear with the order of actions:

“To extend FCS for specific products to an individual’s overall diet, the FCS score for each item reported in a person’s diet was weighted by its percent contribution to that person’s total energy intake, and then summed, to calculate an individual’s Food Compass Score (i.FCS).”

Line 164: Consider to remove duplicated sentence on alcohol.

We believe it is important to mention exclusion of energy contribution from alcohol in the i.FCS calculation, and the inclusion of alcohol intake (as percentage of average total energy intake) as a covariate, as two separate points. We’ve updated the text in the i.FCS paragraph to read:

“Because alcohol is not scored by FCS, energy from alcohol intake was excluded from the i.FCS calculation and included as a covariate in all models.”

Line 210: Suggest to include: “In cross-sectional analysis, all models were adjusted ...” We’ve update this part of the text with slightly different language

Line 232: typo casual. Good find. We’ve updated the text to say “causal”

Line 260: Suggest “With increasing i.FCS, the percentage of participants with Asian or other ethnicity increased; whereas the reverse was observed for Non-Hispanic Black participants.”

Thanks for the suggestion to make this statement more consistent with how we report on other sociodemographic variables. We’ve modified your suggested text slightly in the findings:

“In crude (unadjusted) analyses, adults with a higher i.FCS were more likely to be of Asian or other racial descent and less likely to be of Non-Hispanic Black race/ethnicity;”

Line 270: Suggest to add “...highly correlated with the HEI-2015, ranging from ...”

Noted. We’ve added in the range of spearman correlations for population sub-groups: *“ranging from 0.76 to 0.83”*.

Line 274: Suggest to change to: “In multi ..., higher i.FCS were significantly associated with more favorable risk factors (Table 2).”

We have removed “levels” in the first half of the sentence. We believe it is important to clarify the directionality of the association – that higher i.FCS is associated with lower levels of risk factors (BMI, HBA1c, LDL, TC, BP, etc.) but higher levels of protective factors (i.e., HDL).

Line 297: To make the difference with the survival analysis clearer, suggest to change to: “...was associated with 15% lower odds of having the metabolic syndrome, ...etc”

We have the updated the text to use “prevalence” instead of “risk”, to highlight the cross-sectional nature of these analyses.

Line 343: ‘this NPS’ refers to the FCS.

We have changed the text to say “Food Compass” now instead.

Line 387: Check location of brackets. Good find. We've removed the extra comma inside the bracket.

Line 401: Would be good to explain more, to avoid misinterpretation of black box. We have removed the words "black box" to avoid confusion. We believe the text is clear without that descriptor:

"in comparison to strategies using isolated nutrients or ingredients, such as the FOP warning labels recently implemented in Chile, Mexico, Uruguay, and Israel."

Line 416: Suggest to add "using 1-2 24-hour diet recalls per individual ...". We've modified it to *"up to two 24-hour recalls per individual"*

Table S4: The numbers after the decimal for kcal could be removed. Good suggestion. We have removed all numbers after the decimal point for this Table so that total energy intake is rounded to the kilocalorie.

REVIEWER COMMENTS

Reviewer #1 (Remarks to the Author):

“Validation of the Food Compass Score with a healthy diet, cardiometabolic health, and mortality among U.S. adults, 1999-2018.” O’Hearn et al.

July 11, 2022, Response to June 10, 2022 Response to Reviewer Comments. Mike Rayner, University of Oxford, UK

Re: ‘As recommended by the Reviewer, we now state in the Discussion that neither we nor others have compared the validity of different NPS against health outcomes’ (p3) This is not true. There has been one study (at least) comparing the predictive validity of several NPSs i.e. Egnell M, Seconda L, Neal B, Mhurchu CN, Rayner M, Jones A, Touvier M, Kesse-Guyot E, Hercberg S, Julia C. Prospective associations of the original Food Standards Agency nutrient profiling system and three variants with weight gain, overweight and obesity risk: results from the French NutriNet-Santé cohort. Br J Nutr. 2021 Apr 28;125(8):902-914. doi: 10.1017/S0007114520003384. Epub 2020 Sep 3. PMID: 32878658.

Re: ‘We have added further comments on these important points in the Discussion.’ (p7) Thank you for doing so.

Re ‘The combination of its attributes and domains, together with scoring per 100 kcal, does allow Food Compass to have a consistent and universal algorithm, per definition, as the same algorithm is applied across all food, beverage, and meal product categories.’ (p 8) I agree that Food Compass algorithm is different from that of Nutri-Score , etc. in being applied across all foods rather than by category, but that is not what I meant by the Food Compass algorithm providing no more consistent scoring or convergent validity than other nutrient profile models. But the authors and I are clearly not going to agree on this point so I am willing to let the matter drop.

Re ‘We have added a sentence to the Discussion’ (p9) Thank you for doing this but the sentence does not respond to my original request to the authors to explain their choice of method for converting nutrient profile scores for foods into diet quality scores for individuals. Instead they repeat some of their argument for why a NPS itself should have a 100 kcal reference amount rather than a 100 g reference amount. Their argument is already explored in some depth in the original article describing the Food Compass NPS published in Nature Food (p 816) and need not be repeated here (and anyway the

argument as summarised here is contentious). So I recommend deleting these new sentences beginning, "Both the FCS and i.FCS were weighted by energy..." and ending, 'creating a flawed comparison of the total nutrients provided by each 100g portion.'

Re 'We have clarified the statement: "While some prior NPS validation studies utilized detailed product information from 24 hour recalls,46-51 several utilized food frequency questionnaires 52-55'. (p9) Thank you for doing this.

Re 'The iFCS quintiles appear to be strongly inversely related to total energy intake and there is a weighting for each food stuff on the basis of their contribution to total energy and it appears at least in some of the analyses that TEI was also included as a covariate. A clear statement about what role TEI plays in this score would be helpful.'(p 11)

I agree with Reviewer 2 here but there is an important distinction to be made between the TEI of the diet and the TEI of the food. I am not sure that the authors have understood the point Reviewer 2 is making.

Re.'Line 417:Even though the foods in the food composition table can be precisely categorized, the i.FCS may still suffer from systematic error due to misreporting and omission of food items by participants. Could the authors refer to the implications for this in the section on limitations?...'' could bias results in unpredictable directions." '(p21)

Surely under-reporting of unhealthy items is more likely than over-reporting of healthy items and there is then some predictability in the way reporting will affect the results.?

Re. 'Comparing the nutrients provided by weight of these two foods, pork fat of course provides more of many nutrients per 100g given that it contains no water, leading to absurd claims that pork fat is the 7th healthiest food in the world' (p 22). This statement makes its way into the revised text at line 366 (omitting the word "absurd").

Of course, the claim seems absurd, but any reference amount (100g, 100kJ or serving) will generate apparent absurdities. E.g. NPSs with a 100kJ reference amount generally classify lettuce as an unhealthy food which also seems absurd. I do not think picking out particular foods helps in discussion about the optimal reference amount for an NPS and I would recommend removing the newly added sentences beginning, "Both the FCS and i.FCS were weighted by energy..." and ending, 'creating a flawed comparison of the total nutrients provided by each 100g portion.' (lines 364-369) for this reason in addition to the one I give above.

Reviewer #2 (Remarks to the Author):

The authors present a good response to the issues raised and have made appropriate changes to their manuscript, which is now much improved.

Reviewer #3 (Remarks to the Author):

I thank the authors for answering my questions so thoroughly, clarifying the text and performing the additional analysis. I have no further concerns, only a suggestion to include two subgroup analyses in the new Table S10 by BMI categories and energy reporting status. Where -based on formulae such as the Goldberg equations- the authors could show the associations among low energy reporters, acceptable energy reporters and high energy reporters. This may illustrate the bias due to energy reporting status.

Tufts University Friedman School of Nutrition Science and Policy
150 Harrison Avenue
Boston, MA 02111

August 3, 2022

Response to Reviewer Comments

Thank you to the reviewers for their additional thoughtful feedback on our original submission, **“Validation of the Food Compass Score with a healthy diet, cardiometabolic health, and mortality among U.S. adults, 1999-2018.”** We are pleased to submit our revised manuscript, with careful changes in response to these comments. Our detailed responses to each comment are summarized below.

We believe the manuscript is greatly strengthened as a result of these suggested revisions and analytical updates, and believe this work is now suitable for publication. We look forward your review.

Best regards,

Meghan O’Hearn, MS

PhD Candidate, Friedman School of Nutrition Science and Policy, Tufts University

Meghan.o_hearn@tufts.edu

617-417-6411

REVIEWER COMMENTS

Reviewer #1 (Remarks to the Author):

July 11, 2022, Response to June 10, 2022 Response to Reviewer Comments. Mike Rayner, University of Oxford, UK

Re. 'As recommended by the Reviewer, we now state in the Discussion that neither we nor others have compared the validity of different NPS against health outcomes' (p3) This is not true. There has been one study (at least) comparing the predictive validity of several NPSs i.e. Egnell M, Seconda L, Neal B, Mhurchu CN, Rayner M, Jones A, Touvier M, Kesse-Guyot E, Hercberg S, Julia C. Prospective associations of the original Food Standards Agency nutrient profiling system and three variants with weight gain, overweight and obesity risk: results from the French NutriNet-Santé cohort. Br J Nutr. 2021 Apr 28;125(8):902-914. doi: 10.1017/S0007114520003384. Epub 2020 Sep 3. PMID: 32878658.

Thank you for sharing this paper. Egnell et al. 2021 compared different variations of the same NPS, not different nutrient profiling algorithms. Also, we note that even this paper is a later publication, after other validation papers had been published assessing this sole NPS against health outcomes. All other identified studies similarly only evaluated a single NPS in relation to health outcomes: see Table 1 from our previous Response for examples.

While we believe it was still accurate, we have deleted the statement that "neither we nor others have compared the validity of different NPS against health outcomes." As previously described, we hope to conduct such complex, complementary analyses in future papers, both assessing variations of the Food Compass and comparing other, distinct NPS. We have updated the discussion to further clarify this:

"Additionally, we have not compared the validity of different NPS against health outcomes, and in different nations and subpopulations – an important area for future work."

"Finally, we did not compare the findings to other NPS, which can be done in future work."
(new to Limitation section)

Re 'The combination of its attributes and domains, together with scoring per 100 kcal, does allow Food Compass to have a consistent and universal algorithm, per definition, as the same algorithm is applied across all food, beverage, and meal product categories.' (p 8) I agree that Food Compass algorithm is different from that of Nutri-Score, etc. in being applied across all foods rather than by category, but that is not what I meant by the Food Compass algorithm providing no more consistent scoring or convergent validity than other nutrient profile models. But the authors and I are clearly not going to agree on this point so I am willing to let the matter drop.

Thank you for your input. We agree there are different scientific perspectives on similar issues.

Re 'We have added a sentence to the Discussion' (p9) Thank you for doing this but the sentence does not respond to my original request to the authors to explain their choice of method for converting nutrient profile scores for foods into diet quality scores for individuals. Instead they repeat some of their argument for why a NPS itself should have a 100 kcal reference amount rather than a 100 g reference amount. Their argument is already explored in some depth in the original article describing the Food Compass NPS published in Nature Food (p 816) and need not be repeated here (and anyway the argument as summarised here is contentious). So I recommend deleting these new sentences beginning,

“Both the FCS and i.FCS were weighted by energy...’ and ending, ‘creating a flawed comparison of the total nutrients provided by each 100g portion.’

Sorry if we did not fully understand your previous comment: we did aim to be fully and accurately responsive. See below for further details on how we have modified this section, and removed the pork fat example as you have suggested.

Re ‘The iFCS quintiles appear to be strongly inversely related to total energy intake and there is a weighting for each food stuff on the basis of their contribution to total energy and it appears at least in some of the analyses that TEI was also included as a covariate. A clear statement about what role TEI plays in this score would be helpful.’(p 11)

I agree with Reviewer 2 here but there is an important distinction to be made between the TEI of the diet and the TEI of the food. I am not sure that the authors have understood the point Reviewer 2 is making.

Thank you for this point. It’s relevant to point out that the inverse association between i.FCS quintiles and total energy intake is crude (i.e., not adjusted for other critical sociodemographic variables) and thus should be interpreted with caution. We tested and found, for example, that this association was attenuated when accounting for age and sex:

“However, after adjusting these crude associations by age and sex, the differences by i.FCS quintiles in total energy intake and physical activity were attenuated.”

Total energy intake plays no role in the i.FCS score, and also does not appear to confound or mediate the association when adjusted for as a covariate. We have clarified this further in the methods text: *“The relative energy contribution of each food item was accounted for in the dietary i.FCS calculation. Total energy intake was not included as a component of either the FCS (food level) or the i.FCS (dietary level). Thus, we adjusted for total energy intake derived from the 24-hour recall data as a potential confounder in the models.”*

Re.’Line 417:Even though the foods in the food composition table can be precisely categorized, the i.FCS may still suffer from systematic error due to misreporting and omission of food items by participants. Could the authors refer to the implications for this in the section on limitations?...’ could bias results in unpredictable directions.” ‘ (p21)

Surely under-reporting of unhealthy items is more likely than over-reporting of healthy items and there is then some predictability in the way reporting will affect the results.?

This is interesting thought exercise. We are not aware of clear evidence for a meaningful difference in misreporting “healthful” versus “less healthful” food items that is also differential by risk of poor health outcomes. (Both are necessary: i.e., systematic error only results when the error in the exposure varies according to risk of the outcome). Without such under- or over-reporting also varying according to a person’s risk of disease, the bias would be in unpredictable directions. The most plausible scenario for systematic error would be where individuals at higher health risk would be more likely to under-report unhealthy foods. In this case, this would artificially inflate these individuals’ Food Compass Scores (i.FCS) – a bias which would weaken the reported findings, causing i.FCS to appear less protective than it actually is. Thus, correction for such bias would strengthen our findings. We have updated our Limitations section accordingly:

“If systematic with respect to the outcomes, for instance if individuals at higher health risk were more

likely to underreport unhealthy foods and beverages, this would artificially inflate these individuals' i.FCS – a bias which would weaken the reported findings and make the i.FCS appear less protective than it actually is.”

Re. ‘Comparing the nutrients provided by weight of these two foods, pork fat of course provides more of many nutrients per 100g given that it contains no water, leading to absurd claims that pork fat is the 7th healthiest food in the world’ (p 22). This statement makes its way into the revised text at line 366 (omitting the word “absurd”).

Of course, the claim seems absurd, but any reference amount (100g, 100kJ or serving) will generate apparent absurdities. E.g. NPSs with a 100kJ reference amount generally classify lettuce as an unhealthy food which also seems absurd. I do not think picking out particular foods helps in discussion about the optimal reference amount for an NPS and I would recommend removing the newly added sentences beginning, “Both the FCS and i.FCS were weighted by energy...” and ending, ‘creating a flawed comparison of the total nutrients provided by each 100g portion.’ (lines 364-369) for this reason in addition to the one I give above.

Your example of lettuce is an excellent illustration.

First, a major reason that other NPS create absurd scores is because of the 100g scale. This is a major reason why NPS like Nutri-Score and Health Star Rating must select and use different algorithms and scoring cut-points for different food categories.

Second, your example of lettuce supports our argument, as using a 100kcal (or 100kJ) reference correctly classifies lettuce as a *healthy* foods. In fact, almost all green leafy vegetables (romaine lettuce, collards, cress, chard, kale, spinach, etc.) score 100 when using Food Compass’s (per 100 kcal) algorithm – the highest possible rank in our system. Please see the supplementary materials of our recent Nature Food publication (Table S7, pg 18) for the scoring of all 8032 foods.

<https://www.nature.com/articles/s43016-021-00381-y#Sec14>

Nonetheless, your point is well taken about the danger of picking out particular food scores in any NPS, which is not crucial for our point, and we have thus removed the lines you have suggested.

We have updated the strengths paragraph as follows:

“Both the FCS and i.FCS were weighted by energy (kcal), rather than weight or portion size, which prevents bias or the need for differing scoring cut-points for different food and beverage categories due to differences in water weight, fiber, or fat content.”

Reviewer #2 (Remarks to the Author):

The authors present a good response to the issues raised and have made appropriate changes to their manuscript, which is now much improved.

Thank you for these positive comments. We agree this manuscript is strengthened as a result of your input and suggested revisions.

Reviewer #3 (Remarks to the Author):

I thank the authors for answering my questions so thoroughly, clarifying the text and performing the additional analysis. I have no further concerns, only a suggestion to include two subgroup analyses in the new Table S10 by BMI categories and energy reporting status. Where -based on formulae such as the Goldberg equations- the authors could show the associations among low energy reporters, acceptable energy reporters and high energy reporters. This may illustrate the bias due to energy reporting status.

Thank you for your thoughtful suggestions – we agree that the manuscript is improved as a result of these additional analyses and clarifying text.

We have run the two additional sub-group analyses and added these to the manuscript (see Table below). We have also updated the methods and results text to reflect these new analyses:

“In exploratory analyses, we investigated the relationship between i.FCS and total mortality in subgroups by age, sex, race/ethnicity, education, income, BMI, and plausibility of energy reporting to assess potential variation (interaction) in the association according to these key sociodemographic factors.”

“In exploratory analyses, we investigated whether the relationship between i.FCS and total mortality varied in population subgroups according to age, sex, race/ethnicity, education, income, BMI, and plausibility of energy reporting. Findings were similar across subgroups, with no significant differences in the observed protective associations between i.FCS and mortality (p -interaction >0.05 each) (Table S10).”

Table S10. Multivariable adjusted, survey-weighted prospective association between individual Food Compass Score (i.FCS) and all-cause mortality, stratified by key sociodemographic factors, among U.S. adults, 1990-2018

Sociodemographic factor	N	Events (No. of deaths)	i.FCS, per SD (10.9 points) *
			HR (95% CI)†
Sex			
Male	22996	4158	0.93 (0.88, 0.98)
Female	24992	3323	0.93 (0.88, 0.98)
Age category, years			
20-64	36097	2222	0.90 (0.84, 0.96)
≥65	11839	5259	0.94 (0.89, 0.98)
Race/ethnicity‡			
Hispanic American	12311	1270	0.88 (0.78, 0.98)

Non-Hispanic White	21569	4512	0.93 (0.89, 0.98)
Non-Hispanic Black	9938	1462	0.95 (0.89, 1.02)
Education level			
≤HS graduate	23882	4841	0.94 (0.89, 0.99)
>HS graduate	24036	2651	0.90 (0.85, 0.95)
Income, PIR§			
Low income (PIR ≤2.16)	23832	4370	0.91 (0.86, 0.96)
High income (PIR >2.16)	24086	3111	0.93 (0.89, 0.98)
BMI, kg/m²			
Normal or underweight (<25)	14124	2333	0.89 (0.84, 0.94)
Overweight (25-29)	17569	2636	0.94 (0.89, 1.00)
Obese (BMI ≥ 30)	16225	2512	0.94 (0.88, 1.01)
Energy reporting, TEI:BMR 			
Under-report	12039	2105	0.94 (0.87, 1.01)
Acceptable	34568	4927	0.90 (0.87, 0.94)

* Individual Food Compass Score (i.FCS) calculated as the energy-weighted mean of FCS of all foods consumed, as reported in up to two 24 hour dietary recalls per person, with a potential range from 1 to 100.

† Survey-weighted, multivariable-adjusted Cox proportional hazard model incorporated NHANES dietary recall sample weights to account for the complex survey design and response rates and provide nationally representative effect estimates and associated 95% confidence interval for the non-institutionalized U.S. population. All models adjusted for: age (years), age² (years), sex, race/ethnicity (Mexican-American, other Hispanic, non-Hispanic White, non-Hispanic Black, Asian/other race), education level (<HS graduate, HS graduate, some college or associates degree, ≥ college graduate), income (poverty: income ratio) unless stratifying for that variable; and smoking status (non-smoker, former smoker, current smoker); total physical activity (MET-hours/week), alcohol use (%energy), and self-reported diabetes (yes, no).

‡ Mexican-American and Other Hispanic combined, and Asian/other excluded, from stratified analysis given small sample size and number of events.

§ Represents the ratio of family income to the federal poverty threshold, adjusted for household size. A higher ratio indicates a higher level of income. Individuals were grouped into low and high income, with the cut-point drawn at the median PIR (2.160).

|| TEI:BMR, the ratio of average total energy intake (TEI, in kcal/day) to basal metabolic rate (BMR, kcal/day) was used to determine the plausibility of total energy reporting based on Goldberg cut-offs. BMR was estimated via the Schofield sex- and age-specific predictive equations based on weight and height. TEI was calculated based on the total number of recall days available for each respondent. For example, when two days of data were available, an arithmetic average TEI was calculated. Under-reporters, acceptable reporters and over-reporters were defined as having EI:BMR<0.96, 0.96–2.49 and >2.49 for 2-day dietary recall data, and <0.87, 0.87–2.75 and >2.75 for 1-day dietary recall data, respectively, as previously derived in NHANES.¹ Over-reporters were excluded given the very small sample size and number of events.

AA, Associates Degree; BMI, body mass index; BMR, basal metabolic rate; CI, confidence interval; HS, high school; i.FCS, individual, dietary Food Compass Score; PIR, poverty: income ratio; SD, standard deviation; TEI, total energy intake

References

1. Murakami K, Livingstone MB. Prevalence and characteristics of misreporting of energy intake in US adults: NHANES 2003-2012. *Br J Nutr.* 2015;114(8):1294-1303.

REVIEWER COMMENTS

Reviewer #1 (Remarks to the Author):

Mike Rayner, University of Oxford, 12th August 2022

I have no further comments to make on this manuscript. The authors have addressed my previous comments to my satisfaction.

Reviewer #3 (Remarks to the Author):

Thank you for these additions. It is good to see these results which may also illustrate your text: “If systematic with respect to the outcomes, for instance if individuals at higher health risk were more likely to underreport unhealthy foods and beverages, this would artificially inflate these individuals’ i.FCS – a bias which would weaken the reported findings and make the i.FCS appear less protective than it actually is.”

I would however like to double check below text with the authors, it concerns the footnote in the table: “TEI:BMR, the ratio of average total energy intake (TEI, in kcal/day) to basal metabolic rate (BMR, kcal/day) was used to determine the plausibility of total energy reporting based on Goldberg cut-offs. BMR was estimated via the Schofield sex- and age-specific predictive equations based on weight and height. TEI was calculated based on the total number of recall days available for each respondent. For example, when two days of data were available, an arithmetic average TEI was calculated. Under-reporters, acceptable reporters and over-reporters were defined as having EI:BMR <0.96, 0.96–2.49 and >2.49 for 2-day dietary recall data, and <0.87, 0.87–2.75 and >2.75 for 1-day dietary recall data, respectively, as previously derived in NHANES.¹ Over-reporters were excluded given the very small sample size and number of events.” Murakami et al mention exactly the same cutoffs in their method section: “Consequently, under-reporters, acceptable reporters and over-reporters were defined as having EI:BMR <0.96, 0.96–2.49 and >2.49 for 2-d data and <0.87, 0.87–2.75 and >2.75 for 1-d data, respectively.”

Although the sample is partly shared with the referenced paper (and the cut-off points therefore likely very similar), the current study includes twice as many participants, with their individual weight, height, age, sex and energy intake leading to an individualised BMR and potentially a shift in the mean and 95% cutoffs of TEI:BMR. Did the authors re-calculate the BMR and S for the participants that they have included?

Below is a suggestion:

The differences in N in table 1 have been reduced from ~10000 to ~100. It seems unnecessary to have these minimal differences emphasised so much in a main table. Consider moving Table S4 (population description by quintile i.FCS) to the main manuscript instead (and it would be helpful to include BMI and reporting status in that table) and the current Table 1 to the supplement.

Tufts University Friedman School of Nutrition Science and Policy
150 Harrison Avenue
Boston, MA 02111

August 16, 2022

Response to Reviewer Comments

Thank you to the reviewers for their additional thoughtful feedback on our original submission, **“Validation of the Food Compass Score with a healthy diet, cardiometabolic health, and mortality among U.S. adults, 1999-2018.”** We are pleased to submit our revised manuscript, with careful changes in response to these comments. Our detailed responses to each comment are summarized below.

We believe the manuscript is now suitable for publication. We look forward your review.

Best regards,

Meghan O’Hearn, MS

PhD Candidate, Friedman School of Nutrition Science and Policy, Tufts University

Meghan.o_hearn@tufts.edu

617-417-6411

REVIEWER COMMENTS

Reviewer #1 (Remarks to the Author):

Mike Rayner, University of Oxford, 12th August 2022

I have no further comments to make on this manuscript. The authors have addressed my previous comments to my satisfaction.

Reviewer #3 (Remarks to the Author):

Thank you for these additions. It is good to see these results which may also illustrate your text: “If systematic with respect to the outcomes, for instance if individuals at higher health risk were more likely to underreport unhealthy foods and beverages, this would artificially inflate these individuals’ i.FCS – a bias which would weaken the reported findings and make the i.FCS appear less protective than it actually is.”

I would however like to double check below text with the authors, it concerns the footnote in the table: “TEI:BMR, the ratio of average total energy intake (TEI, in kcal/day) to basal metabolic rate (BMR, kcal/day) was used to determine the plausibility of total energy reporting based on Goldberg cut-offs. BMR was estimated via the Schofield sex- and age-specific predictive equations based on weight and height. TEI was calculated based on the total number of recall days available for each respondent. For example, when two days of data were available, an arithmetic average TEI was calculated. Under-reporters, acceptable reporters and over-reporters were defined as having EI:BMR <0.96, 0.96–2.49 and >2.49 for 2-day dietary recall data, and <0.87, 0.87–2.75 and >2.75 for 1-day dietary recall data, respectively, as previously derived in NHANES.1 Over-reporters were excluded given the very small sample size and number of events.”

Murakami et al mention exactly the same cutoffs in their method section: “Consequently, under-reporters, acceptable reporters and over-reporters were defined as having EI:BMR <0.96, 0.96–2.49 and >2.49 for 2-d data and <0.87, 0.87–2.75 and >2.75 for 1-d data, respectively.”

Although the sample is partly shared with the referenced paper (and the cut-off points therefore likely very similar), the current study includes twice as many participants, with their individual weight, height, age, sex and energy intake leading to an individualised BMR and potentially a shift in the mean and 95% cutoffs of TEI:BMR. Did the authors re-calculate the BMR and S for the participants that they have included?

Since Murakami *et al.*'s cut-points were based on a nationally representative NHANES sample as well, we did not feel 95% cut-points would vary substantially, and so for consistency used the same cut-points as in their published paper.

Below is a suggestion:

The differences in N in table 1 have been reduced from ~10000 to ~100. It seems unnecessary to have these minimal differences emphasised so much in a main table. Consider moving Table S4 (population description by quintile i.FCS) to the main manuscript instead (and it would be helpful to include BMI and reporting status in that table) and the current Table 1 to the supplement.

We agree that, with the additional available data on the prospective mortality analyses, the differences in the two columns are trivial, and so we have deleted the second column. We have retained Table S4 as a supplement given that these are exploratory, crude (unadjusted) associations.

We have added the following explanation to Table 1 footnote:

“81 fewer subjects were included in the prospective mortality analyses due to lack of linkage to the National Death Index.”

Per your suggestion, we have also included the n (%) of NHANES respondents by energy reporting status (underreport, acceptable, and overreport, based on TEI:BMR) in Table 1 with an explanatory footnote.

REVIEWERS' COMMENTS

Reviewer #3 (Remarks to the Author):

I thank the authors for their reflection. I have no further comments.